# Explaining the period fluctuation of the quasi-biennial oscillation

Young-Ha Kim[1]

[1]Research Institute of Basic Sciences, Seoul National University, Seoul, Republic of Korea

**Correspondence:** Young-Ha Kim (kyha0830@snu.ac.kr)

**Abstract.** The tropical stratosphere is characterized by a periodic oscillation of wind direction between westerly and easterly, known as the quasi-biennial oscillation, which modulates middle atmospheric circulations and surface climate on interannual time scales. The oscillation period fluctuates irregularly between 20 and 35 months. The causes of this fluctuation have long been hypothesized but lack observational evidence. This study shows that the period fluctuation is primarily driven by variability in small-scale wave (gravity wave) activity, with an additional contribution from variability in tropical upwelling. Using an atmospheric reanalysis dataset, we capture temporal variations in small-scale wave activity that are coherent with the varying speed of the oscillation. This wave activity variation stems from the seasonality of tropical convection and tropopause-layer wind, revealing their fundamental role in modulating the quasi-biennial period. Our findings suggest that better representing these multi-scale interactions in models can enhance the accuracy of seasonal forecasts and the reliability of future climate projections.

## 1 Introduction

The wind in the tropical stratosphere alternates between an easterly and westerly direction with periods varying irregularly from 20 to 35 months (Baldwin et al., 2001; Anstey et al., 2022a). This so-called quasi-biennial oscillation (QBO) is intense enough in amplitude ($\sim 30 \ \mathrm{m \, s^{-1}}$) to modulate the extratropical circulation and global distributions of chemical species (e.g., Holton and Tan, 1980; Ling and London, 1986; Randel and Wu, 1996; Kinnersley and Tung, 1998; Flury et al., 2013). The alternating wind phases propagate down to the tropopause layer ($\sim 18 \ \mathrm{km}$ altitude). Recent studies have revealed further downward impacts of the QBO on surface weather and climate in the tropics (Haynes et al., 2021; Yoo and Son, 2016) and extratropics through atmospheric teleconnections (Thompson et al., 2002; Garfinkel and Hartmann, 2011; Gray et al., 2018; Anstey et al., 2022b). The QBO is therefore regarded as an important climate mode that can enhance the skill of seasonal and decadal predictions, provided that it is reproduced well by forecasting models (Scaife et al., 2014; Coy et al., 2022).

A barrier to the successful prediction of the QBO is the variability in its period and phase progression. The speed of phase progression varies, largely depending on the calendar month, and accordingly, the lengths of individual oscillation cycles fluctuate (e.g., Dunkerton, 1990; Wallace et al., 1993; Schenzinger et al., 2017; Coy et al., 2020). Current seasonal prediction models are deficient in capturing the varying timings of phase transition, which limits the forecast skill of the oscillation beyond the lead time of a few months (Stockdale et al., 2022; Coy et al., 2022). In line with this, climate models tend to reveal a lack of variability in the QBO period in multi-decadal simulations (Bushell et al., 2022). Regarding this problem, a fundamental

question is the underlying process driving the observed fluctuations in the period and phase progression speed, which remains unresolved.

The QBO is primarily a dynamic phenomenon driven by momentum forcing due to atmospheric waves (Lindzen and Holton, 1968; Holton and Lindzen, 1972). Observational and theoretical studies have demonstrated that a broad spectrum of atmospheric waves, generated by heat release from tropical convection, propagate vertically into the stratosphere (e.g., Holton, 1972; Salby and Garcia, 1987; Horinouchi and Yoden, 1996; Song et al., 2003; Song and Chun, 2005). They transport and deposit momentum to the flow in the stratosphere around the easterly/westerly jet of the QBO. This acts as the restoring force of the oscillation, advancing its phase (Lindzen and Holton, 1968; Holton and Lindzen, 1972). However, the flow in the tropical stratosphere is not purely horizontal but weakly upward on average (by less than $1 \, \mathrm{mm \, s^{-1}}$) (Schoeberl et al., 2008; Kim and Chun, 2015b). Albeit very slow compared to horizontal motions, this upwelling advects substantial momentum, which can largely offset the wave forcing, thereby retarding the oscillation. Therefore, the theoretical hypothesis suggests that the fluctuation in the QBO progression speed could be explained by variations in the magnitude of upwelling and/or wave activity (e.g., Dunkerton, 1990; Kinnersley and Pawson, 1996; Hampson and Haynes, 2004; Kim et al., 2013; Krismer et al., 2013; Anstey et al., 2022a).

To date, no observational evidence has been found for the coherent variation of the wave activity with the fluctuating QBO periods (for a climate model result, see Krismer et al., 2013). A difficulty has existed in this issue because of the limitation of measurements in spatial resolution and coverage, given the ubiquitous convection over the tropics and the broad wave spectrum down to the ∼10 km scale in horizontal wavelength. On the other hand, the stratospheric upwelling is known to have a seasonal cycle (Mote et al., 1996, 1998; Schoeberl et al., 2008), and its possible effects on the QBO variability have been extensively studied using observational records and modeling approaches (e.g., Kinnersley and Pawson, 1996; Hampson and Haynes, 2004; Rajendran et al., 2018; Coy et al., 2020). In particular, it has been suggested that the seasonal dependence of QBO progression, including its stalling (Wallace et al., 1993; Kinnersley and Pawson, 1996), could result from the seasonal cycle of the upwelling. In addition, the decadal-scale modulation of the QBO period by the solar cycle has also been proposed, potentially linked to the solar effect on global stratospheric circulation, including tropical upwelling (Salby and Callaghan, 2000; Pascoe et al., 2005). However, subsequent observations have shown that the relationship between the solar cycle and the QBO is inconsistent over time and remains uncertain (e.g., Fischer and Tung, 2008).

Here, we use a global reanalysis data to investigate the fluctuations in the QBO period and progression speed. Our study reveals the underlying process driving these fluctuations, providing new evidence of the coherent variation of small-scale wave activity with the QBO periods.

## 2 Data and methods

### 2.1 Datasets

European Centre for Medium-Range Weather Forecasts (ECMWF) Reanalysis v5 (ERA5, Hersbach et al., 2020) data were used throughout the study. ERA5 is a global reanalysis spanning 1940 onward with a spatial resolution of ∼30 km. We used

hourly wind and temperature data over the period of 1956–2015. The analysis period was determined to begin with the first observation of the QBO up to 10 hPa altitude. We excluded years after 2015 because the oscillation phases could not be unambiguously defined due to the QBO disruptions occurred in 2016 and 2019–2020 (e.g., Osprey et al., 2016). The conventional pressure-level dataset was used for all analyses (such as wave momentum fluxes and momentum forcing, described in Sects. 2.3 and 2.4), except for calculating the QBO-phase descent speed and the 84 hPa wave flux (Sect. 4.1). For the descent speed, finer vertical profiles of monthly zonal wind were obtained from the ERA5 native model-level dataset. For the annual cycle of precipitation, Long-Term Mean data from the Global Precipitation Climatology Project (GPCP) Monthly Analysis version 2.3 (Adler et al., 2003) were used.

## 2.2 QBO cycles and periods

Monthly and zonally averaged zonal winds over 5° N–5° S were used to describe the QBO. At each height, we defined an oscillation cycle as the period between the onsets of westerly winds. To avoid noise around zero wind and uniquely detect the onset of westerly winds, the 1-2-1 temporal smoothing was applied three times to the monthly time series, but only when defining the cycle.

## 2.3 Wave momentum fluxes

Upward fluxes of eastward and westward momentum due to waves were calculated monthly, after decomposing waves via Fourier transform in longitude and time. For this 2-dimensional Fourier transform, a 34 d window centered on the target month was used, with sine-tapering applied over 4 d on each side (leaving the central 26 d untapered). The eastward (westward) momentum fluxes were obtained by summing positive (negative) zonal-momentum fluxes in the Fourier domain where the zonal phase speeds were positive (negative). Absolute values were taken for the westward-momentum fluxes.

In this study, we refer to small-scale waves as those with zonal wavenumbers greater than 20 (wavelengths less than ∼2000 km, i.e., gravity waves). When analyzing momentum fluxes of small-scale waves, only waves with zonal phase speeds greater than $\pm 5$ m s$^{-1}$ were considered. Local ambient winds for small-scale waves vary zonally in the tropical upper troposphere (e.g., due to the Walker circulation), allowing for both eastward- and westward-momentum fluxes at a given low phase speed, depending on the location of the waves. The phase-speed threshold ($\pm 5$ m s$^{-1}$) was applied to avoid artificial variability in the calculated momentum fluxes, as fluxes in opposite directions would offset each other in the spectrum at low phase speeds.

## 2.4 Momentum forcing

The change rate of zonally averaged wind is expressed as

$$\frac{\partial \overline{u}}{\partial t} = \frac{\nabla \cdot \boldsymbol{F}}{\rho_0 \, a \cos \phi} - \overline{w}^* \frac{\partial \overline{u}}{\partial z} + \overline{v}^* \hat{f} \tag{1}$$

by defining the Eliassen–Palm (EP) flux $\boldsymbol{F}$ and meridional and vertical velocities of the residual circulation $(\overline{v}^*, \overline{w}^*)$ following Andrews et al. (1987). Here, $\rho_0$, $a$, and $(\phi, z)$ are the reference density, earth radius, and latitudinal and vertical coordinates, respectively; $\hat{f} := f - (a \cos \phi)^{-1} \partial(\overline{u} \cos \phi)/\partial \phi$, where $f$ is the Coriolis parameter. The first two terms on the right-hand side

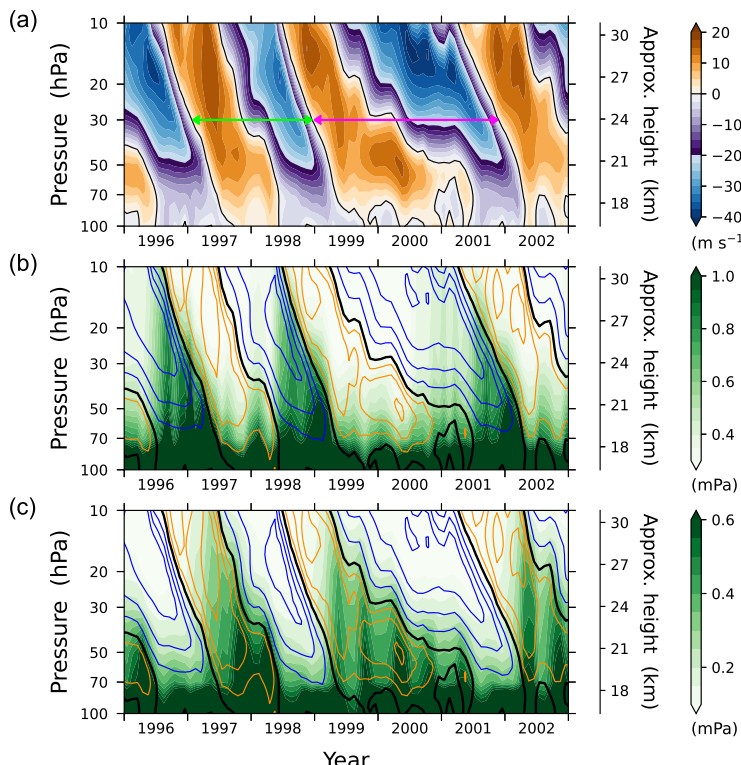

**Figure 1.** (a) Stratospheric zonal wind velocity averaged over 5° N–5° S. The green and magenta arrows indicate two examples of oscillation cycles at 30 hPa altitude, highlighting a large variation in the QBO period. (b, c) Upward fluxes of (b) eastward and (c) westward momentum due to atmospheric waves over 15° N–15° S (shading) along with the wind (yellow contours for westerlies at 5 m s$^{-1}$ intervals; blue contours for easterlies at 10 m s$^{-1}$ intervals; thick black contours for zero winds).

of Eq. (1) represent the major components of momentum forcing in the equatorial stratosphere: total wave forcing (contributed by waves of all scales) and vertical advection by upwelling, respectively. The terms in Eq. (1) were calculated using hourly data and averaged monthly. Additionally, an indirect estimate of the total wave forcing was also obtained by subtracting all terms except the first on the right-hand side from the wind change rate in Eq. (1).

## 3   Characteristic behaviors of the QBO


Figure 1a shows the QBO of the tropical stratospheric wind with alternating phases, highlighting a large variation in its period (horizontal arrows) in the late 1990s–early 2000s (see Fig. 2 for the record of QBO periods since 1956). A characteristic phenomenon associated with the period variation is the change in the descent speed of the easterly phase (Fig. 1a). The downward propagation of this phase often stagnates, as observed in the period from late 2000 through early 2001, resulting in

a lengthening of the oscillation cycle. However, the downward speed of the westerly phase barely changes between cycles or

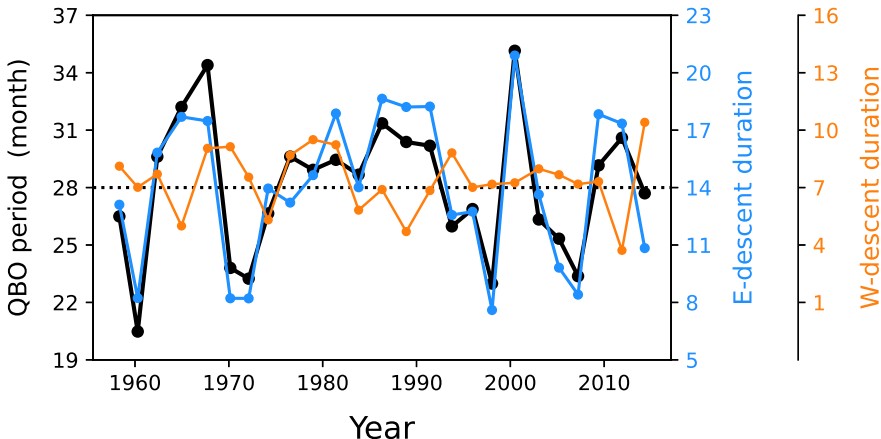

**Figure 2.** QBO period measured at 30 hPa (black), along with the durations of the easterly-phase (blue) and westerly-phase (orange) descent from 15 hPa to 50 hPa, for QBO cycles over 1956–2015. The averages of the three variables over the cycles (28, 14, and 7 months, respectively) are indicated by the dotted horizontal line. Pearson correlation between QBO periods and easterly-phase descent durations is 0.91.

within each cycle, as measured by the slopes of the zero-wind lines at the bounds of cycles in Fig. 1a. This results in the period of each cycle being nearly constant with height.

Another characteristic of the QBO is that the onset of westerly phases at 10 hPa roughly coincides with the end of those phases at about 70 hPa (near the bottom of the tropical stratosphere), and the same pattern holds for easterly phases (Fig. 1a).
This can be explained by the established theory of the interaction between atmospheric waves and flow (Holton and Lindzen, 1972; Plumb, 1977). Figure 1b shows upward fluxes of eastward momentum carried by the waves. The eastward-momentum fluxes are largely reduced under the westerly flow condition, as the theory suggests. Therefore, at an arbitrary altitude, these fluxes can be substantial only when the flow is easterly throughout the layer below. The formation of such an easterly layer by the cessation of the westerly phase at 70 hPa allows eastward momentum to be transported to ∼10 hPa altitude, acting as a
conduit for the stream of momentum. The eastward momentum there forces the flow, leading to the transition to the westerly phase (Fig. 1b). Similarly, the westerly flow condition at the lower altitude works as a conduit for westward momentum (Fig. 1c), thereby controlling the onset of the easterly phase at ∼10 hPa.

Considering the aforementioned behaviors of the QBO, its period can also be estimated by the duration of westerly-phase descent through the layer from ∼10 hPa to 70 hPa plus that of easterly-phase descent. Furthermore, the variation of the period
is predominantly caused by the latter, given that the descent speed of the westerly phase barely changes. To demonstrate this, the periods of the QBO from 1956 to 2015 are shown in Fig. 2, along with the durations of easterly-phase and westerly-phase descents. The easterly-phase descent was measured by tracking a constant zonal-wind velocity ($U_\varphi = -10 \text{ m s}^{-1}$) within the easterly-shear layer between 15 hPa and 50 hPa. (This altitude range was chosen because the phase descent above 15 hPa was often too fast to measure using monthly data, while below 50 hPa, the descent was not clearly identifiable in some years

due to the attenuation of the QBO near the tropopause.) The same approach was applied to the westerly-phase descent, except it tracked $U_\varphi = 5\,\mathrm{m\,s^{-1}}$ (due to the small westerly amplitude; see Fig. 1a) within the westerly-shear layer.

Figure 2 demonstrates that variations in the duration of easterly-phase descent predominantly account for the variations in the period of whole cycle. For every QBO cycle, the period is precisely estimated by the duration of easterly-phase descent through the 15–50 hPa layer plus an offset of 14 months, within an error of less than $\pm 3$ months. This result was not sensitive 125  to the value of $U_\varphi$ for the easterly-phase descent in the range of 0 to $-15\,\mathrm{m\,s^{-1}}$. Additionally, we found that the precision of this period estimation is not further improved by explicitly considering the duration of westerly-phase descent (i.e., summing the durations of both descents through the layer and applying a reduced offset). Therefore, understanding the QBO period fluctuations primarily reduces to uncovering the process that controls the easterly-phase descent.

## 4  Variations in phase progression speed

### 4.1  Regression of phase descent speed


The descent speed of easterly phase, following a constant zonal-wind velocity $U_\varphi$ in easterly shear, can be approximated as

$$-\dot{z}_\varphi = \frac{\partial \overline{u}}{\partial t} \left( \frac{\partial \overline{u}}{\partial z} \right)^{-1} \approx \rho_0^{-1} |\mathcal{F}(U_\varphi)| - \overline{w}^* \tag{2}$$

from Eq. (1), while neglecting meridional gradient of wind near the equator. Here, $\mathcal{F}(c)$ denotes the spectral density of zonal-momentum flux with respect to the zonal phase speed of waves $c$, at the bottom of the stratosphere. Following Lindzen and Holton 135  (1968), we consider the critical-level absorption to be the major wave-dissipation mechanism, so that the EP flux divergence term in Eq. (1) has been approximated as $\rho_0^{-1} |\mathcal{F}(U_\varphi)| \partial \overline{u}/\partial z$. The critical-level absorption, as the primary forcing mechanism of gravity waves for the QBO, has been supported by observations (e.g., Ern et al., 2014).

Motivated by Eq. (2), we linearly regress the descent speed of the easterly phase at each pressure level ($p$) by

$$-\dot{z}_\varphi(p) \approx \alpha(p) F_{84} - \beta(p) W + \delta(p) \tag{3}$$

where $F_{84}$ is the westward-momentum flux due to small-scale waves (see Sect. 2.3) averaged over 15° N–15° S at 84 hPa ($\sim$17 km) altitude, and $W$ is the mass-weighted average of $\overline{w}^*$ ($\int \overline{w}^* \mathrm{d}p / \int \mathrm{d}p$) over the 10–70 hPa layer and over the same latitude band (representing the stratospheric upwelling in the QBO domain), with $\alpha$ and $\beta$ being their regression coefficients. The constant term, $\delta$, denotes the intercept of the regression. Here, the small-scale waves (with horizontal wavelengths less than $\sim$2000 km) were taken into account for $F_{84}$, as these have been inferred to be the primary source of the momentum forcing in 145  the stratosphere during the descending easterly phase (e.g., Ern et al., 2014; Kim and Chun, 2015b). The 84 hPa altitude in the ERA5 native levels was used because momentum flux at 70 hPa can be influenced by the QBO, while the 100 hPa level is too close to the deep convective wave source. The wide latitude band was chosen to account for the lateral propagation of waves toward the equator (Kim et al., 2024). The spatial averaging applied to $\overline{w}^*$ aimed to reduce the effect of the QBO-induced local circulation (for details on this circulation, see Plumb and Bell, 1982). By minimizing QBO-induced factors in the independent

variables of the regression (rather than using flux at a higher altitude or local $\overline{w}^*$), the regressed descent speed can be causally attributed to these variables. Additionally, we note that using local $\overline{w}^*$ did not yield an overall better regression score compared to using $W$, although this analysis is not shown here.

Again, we used $U_\varphi = -10 \ \mathrm{m \, s^{-1}}$ for the analysis, while the results were not sensitive to it in the range of 0 to $-15 \ \mathrm{m \, s^{-1}}$. The regression was based on the monthly mean time series during 1956–2015, for the selected months when the QBO wind phase of $U_\varphi$ appeared at around a given regression-target pressure $p$ with a tolerance of $\pm(1/8)p$. The numbers of samples were 124, 95, 107, 78, and 54, respectively, at 60, 50, 40, 30, and 20 hPa (inversely proportional to the descent speeds at those pressure levels). These numbers correspond to 2–5 per QBO cycle on average. Note that the sampling at the monthly interval is appropriate for providing sufficient degrees of freedom for analysis, given that $F_{84}$ exhibits significant month-to-month variability (i.e., low autocorrelation), as will be shown later. In addition, the correlation between $F_{84}$ and $-W$ was generally small (within $\pm 0.13$ for all regression altitudes), so their linear dependence can be ignored. The selected time series of $F_{84}$, $-W$, and $-\dot{z}_\varphi$ for each altitude were standardized prior to the regression (resulting in zero means and unit standard deviations) to assess the relative contributions of the wave flux and stratospheric upwelling to the descent speed variation.

Figure 3a shows variations in the observed speed of the easterly-phase descent throughout 1956–2015 (the pressure scale height of 6.3 km was used here). The descent speed varies widely, as indicated by its standard deviation being comparable to its mean at each altitude, while it tends to increase with altitude. Figure 3b presents the standardized regression coefficients of $F_{84}$ and $-W$ obtained for the descent speed at each altitude, along with the regression scores. A direct comparison between the observed and regressed speeds is shown in Fig. 4 for several altitudes (20, 30, 40, and 50 hPa). Overall, the regression scores are high (the regression correlations are $\sim 0.8$ at most altitudes), with generally large coefficients of $F_{84}$ (0.5–0.8) (Fig. 3b). The coefficients of $-W$ are small at high altitudes (0.2–0.3 at $p \lesssim 30$ hPa), but increase at lower altitudes, reaching up to 0.6 at 45 hPa. This suggests that wave flux entering the stratosphere dominates descent speed variation at higher altitudes, while at $p \gtrsim 40$ hPa, the contribution of upwelling variability becomes comparable to that of wave flux.

Utilizing the obtained regression coefficients, a hindcast is performed using the climatological-mean (1956–2015) annual cycle of $F_{84}$ and $-W$ as input. The correlation between the hindcasted and observed descent speeds (Fig. 3b, grey dotted line) is very close to the regression correlation. This result indicates that the seasonal variation in wave flux and upwelling accounts for a large portion of the fluctuations in the descent speed of the easterly phase.

To examine the seasonal variation, we present the descent speeds at 20, 30, 40, and 50 hPa as a function of calendar month in Fig. 5, along with $F_{84}$ and $W$ at the corresponding time, i.e., when the easterly phase of $U_\varphi$ descents through each altitude. (Note that an inverted axis is used for $W$.) Individual monthly values (dots) and their averages for each calendar month (solid lines) are plotted. Additionally, the climatological-mean annual cycles of $F_{84}$ and $W$, obtained from the full time series regardless of QBO phase, are superimposed (dashed lines).

The descent speed (black solid line) exhibits a semiannual variation at 20 hPa, with maxima in April–May and October–November. Compared to the former maximum, the latter weakens with decreasing altitude, becoming almost extinct at 50 hPa. Notably, the descent speed is nearly zero around January and August at 20–40 hPa, indicating the stalling of phase descent. At 50 hPa, the descent speed is generally low throughout the period from August to February, while the stalling occurs mainly in

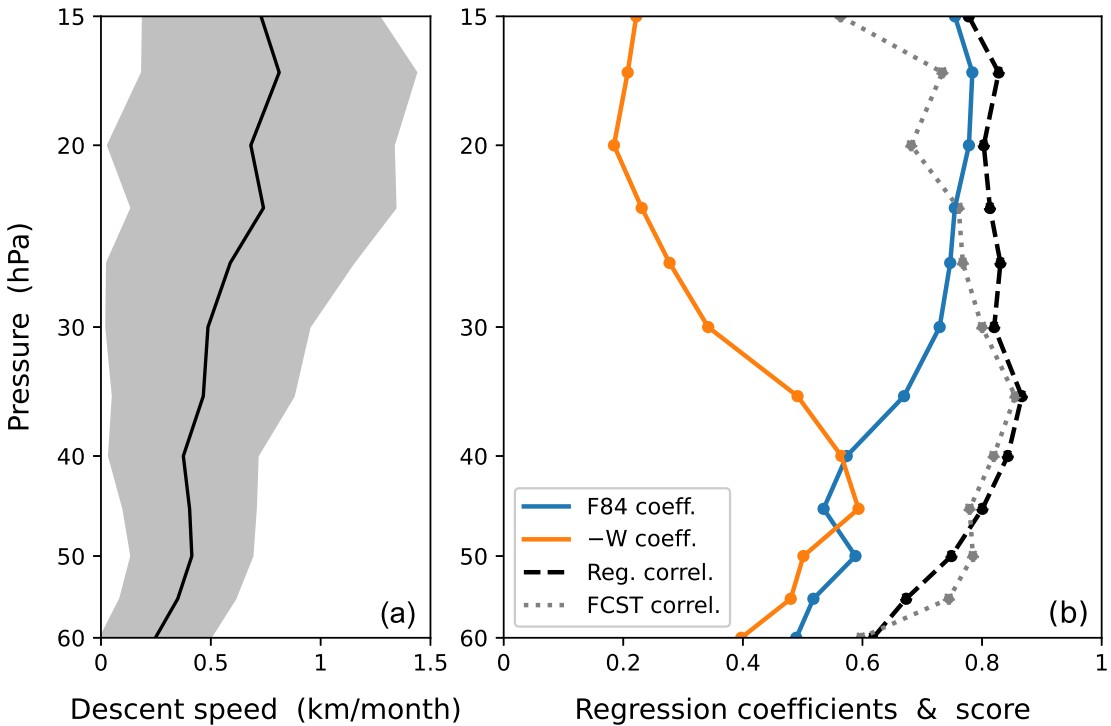

**Figure 3.** (a) Climatological mean (black line) and variation within one standard deviation (grey shading) of the descent speed of the easterly QBO phase (measured following the vertical trajectory of the $-10\,\mathrm{m\,s^{-1}}$ wind phase in easterly shear) throughout 1956–2015. (b) Standardized regression coefficients of 84 hPa small-scale westward-wave momentum flux (blue line) and stratospheric upwelling (orange line) for the descent speed at each altitude, along with the regression correlation score (black dashed line) (see the text for details of the regression). The grey dotted line in (b) depicts the correlation score of a hindcast using the obtained coefficients and the seasonal cycle of momentum flux and upwelling.

January (also see Fig. 1a). The seasonal variation in descent speed shown in Fig. 5 is broadly consistent with previous studies (e.g., Wallace et al., 1993, Fig. 13; Coy et al., 2020, Fig. 1).

The annual cycles of $F_{84}$ and $W$ derived for the easterly-descending phases (blue and orange solid lines, respectively) show similar variations to their climatological-mean cycles (dashed lines) at all altitudes (Fig. 5). $F_{84}$ exhibits a semiannual variation, whereas $W$ is dominated by an annual variation, resulting in their linear independence to a reasonable approximation

(with a correlation of $-0.11$ between the climatological cycles). Figure 5 indicates that the descent speed variation at 20–30 hPa primarily follows the variation in $F_{84}$. However, the influence of $W$ variability increases with decreasing altitude. At 40–50 hPa, the descent speed variation appears to be explained only when both variables are collectively considered. This seasonal-cycle analysis supports the regression results shown in Fig. 3b.

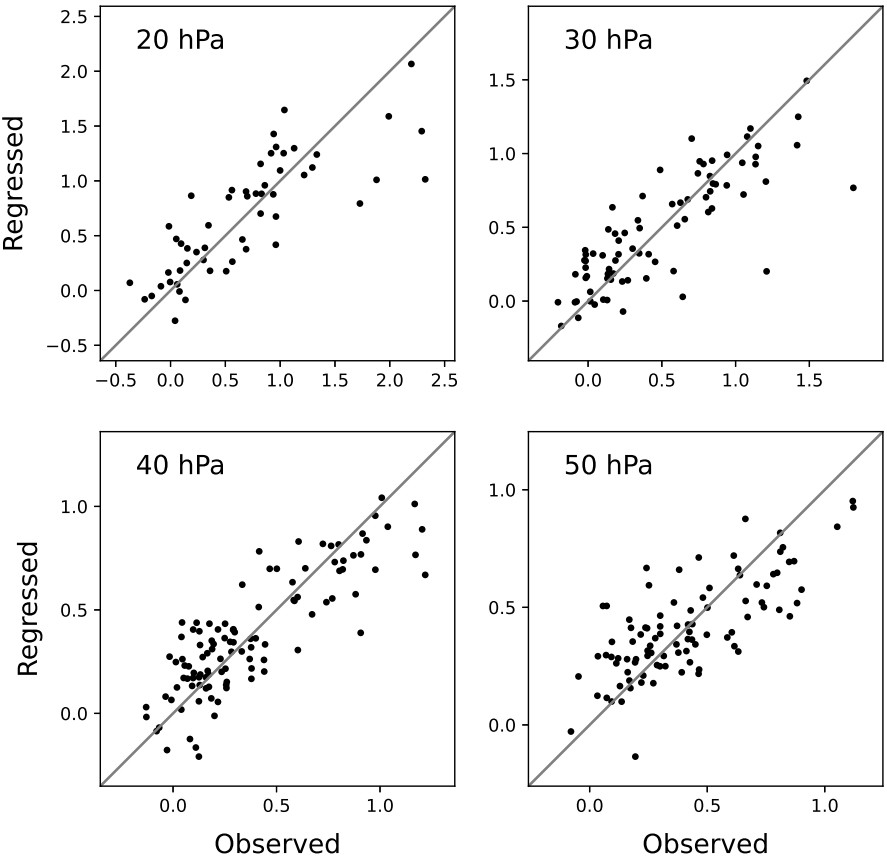

**Figure 4.** Scatter plots of observed versus regressed speeds of the easterly-phase descent (in km per month) at 20, 30, 40, and 50 hPa altitudes.

Additionally, individual values of $F_{84}$, $W$, and descent speed show substantial spread within each calendar month. These variations, on interannual or longer time scales, may not have been successfully modelled in the regression, thereby limiting the regression score to $\sim$0.8 (Fig. 3b). This is inferred from the fact that the hindcast using the climatological annual cycle yields a similar correlation score. Further investigation of these longer-term variations is left for future studies, while the remaining sections concentrate on seasonal variation.

To confirm the limited contribution of larger-scale waves, we also performed the same regression except that $F_{84}$ was replaced with the flux from the westward propagating inertio-gravity mode (with zonal wavenumbers of up to 20 and periods shorter than 2.5 d) (not shown). The regression coefficients of the larger-scale wave flux were much smaller (0.2–0.5 at most altitudes), with worse regression scores (e.g., down to 0.3 at 15 hPa), compared to those using $F_{84}$.

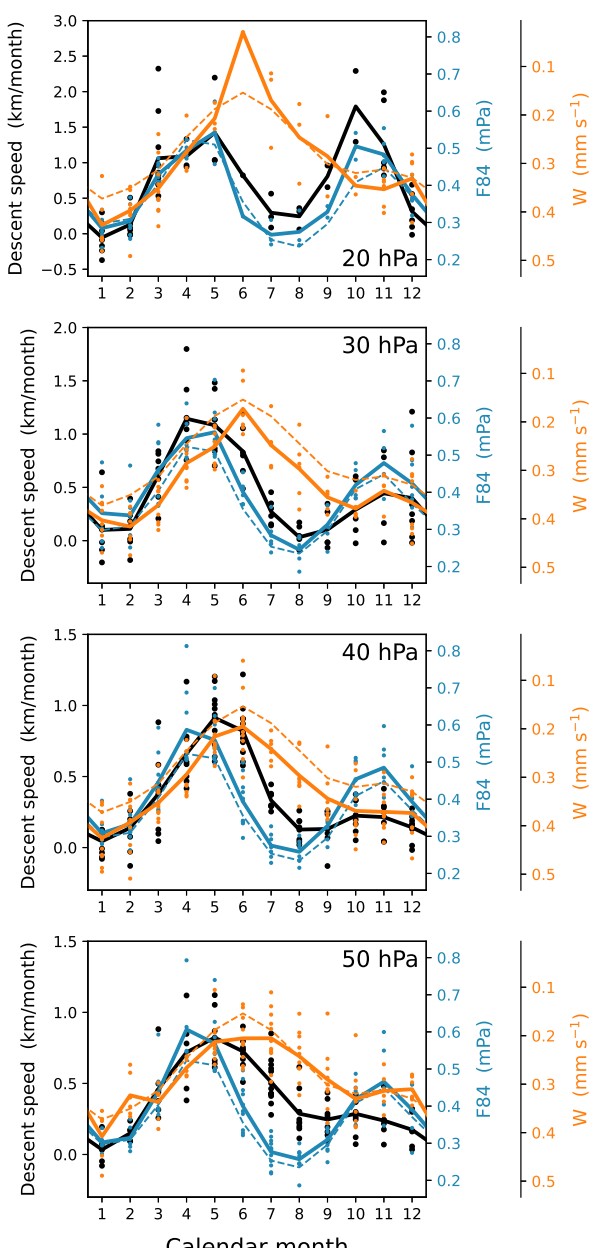

**Figure 5.** Phase-descent speed (black dots), 84 hPa small-scale westward-wave momentum flux (blue dots), and stratospheric upwelling (orange dots, with an inverted axis) as a function of calendar month, during the easterly-descending phases at 20, 30, 40, and 50 hPa (from top to bottom) throughout 1956–2015. Solid lines indicate their averages for each calendar month. Climatological-mean annual cycles of the wave flux and upwelling, derived from the full time series regardless of QBO phase, are additionally plotted in all panels (dashed lines).

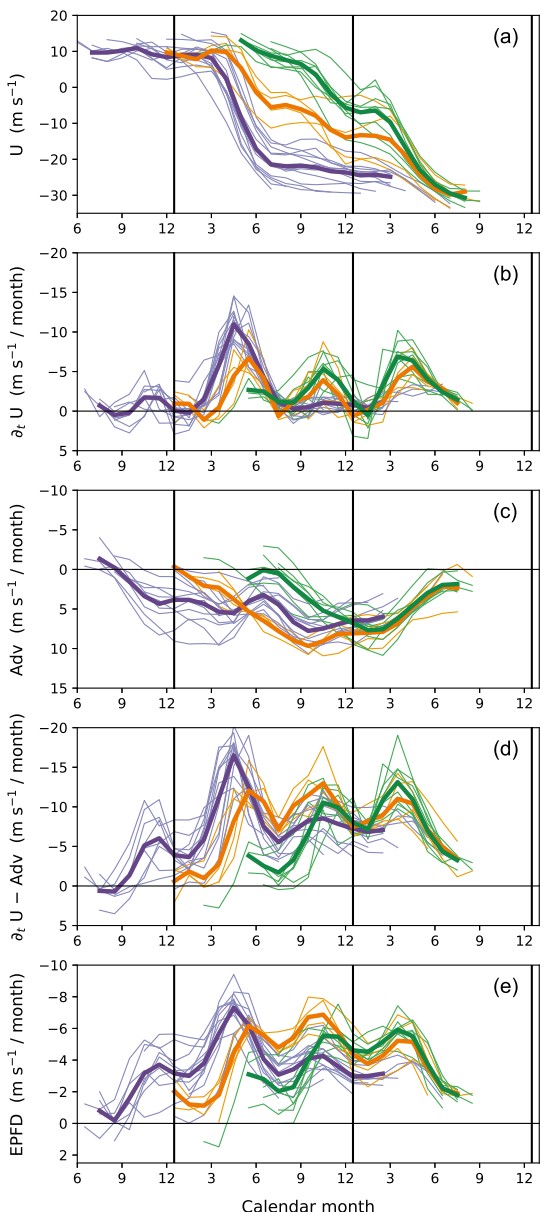

**Figure 6.** Time series at 30 hPa from the westerly-maximum to the easterly-maximum phase in each cycle of the QBO during 1956–2015 (thin lines), showing (a) the zonal wind over 5° N–5° S, (b) its tendency, (c) the sum of advection and Coriolis force (Adv), (d) the tendency with the contribution of Adv subtracted (i.e., (b) minus (c)), and (e) the Eliassen–Palm flux divergence (see Eq. (1)), as a function of calendar month. The cycles are divided into three groups (indicated by different colors) based on the pattern of phase progression (see the text), and the averages for each group are plotted by thick lines. Note that the y-axes in (b)–(d) use the same scale interval, and the scale in (e) is half that.

## 4.2 Temporal variations in momentum forcing

The time evolutions of wind and associated momentum forcing at fixed altitudes are analyzed to investigate the temporal

variations of phase progression speed. Figure 6a and b show the time series of winds at 30 hPa (a middle level of the vertical QBO span) and their change rates, respectively, during the westerly-to-easterly transition phase in every QBO cycle in 1956–2015 (thin lines), as a function of calendar month. In all cycles, the phase progression occurs only during certain months: March to July and September to December, consistent with previous studies (e.g., Schenzinger et al., 2017). Accordingly, the cycles can be divided into three groups based on the pattern of phase progression. The first group (purple lines in Fig. 6) consists

of cycles in which the phase change occurs essentially during March to July. The second group (green lines) includes cycles where the phase changes first to a weak easterly during September to December, stagnates until the following February, and then changes further in March to July. The third group (orange lines) experiences the first phase change in March to July and the final change in next March to July. (See Appendix A for the categorization procedure.) The thick lines in Fig. 6 represent the averaged time series for each group. Note that the stagnation of phase over time at a given altitude results in the stalling of

phase descent (see Fig. 1a; Fig. 5, black line).

Figure 6c presents the contribution of meridional circulation $(\bar{v}^*, \bar{w}^*)$ to the wind tendencies (i.e., sum of the vertical and meridional advection and the Coriolis force in Eq. (1)) at 30 hPa. In general, they have the opposite sign to the observed wind tendencies (cf. Fig. 6b), implying the retarding effect of the upwelling on the phase evolution. However, they show rather slow temporal variations that are not coherent with those of the observed tendencies, for each group of cycles. For quantitative

comparison, Fig. 6d shows the portion of the wind tendencies with the contribution of meridional circulation subtracted (i.e., the tendencies minus all terms except the EP flux divergence in Eq. (1)). These time series still exhibit large monthly variations similar to the observed tendencies (Fig. 6b), with the two peaks in March to July and September to December, respectively. Note that, given the momentum budget (Eq. (1)), the time series shown in Fig. 6d are regarded as an indirect estimate of the total wave forcing, up to potential errors in upwelling velocity in the reanalysis data.

The direct estimate of the wave forcing, as resolved in the reanalysis field (i.e., the explicitly calculated EP flux divergence), is presented in Fig. 6e. Its time series exhibit very similar temporal variations to the indirect estimate of the forcing (Fig. 6d) throughout the phase evolution: both estimates of the forcing start increasing in the early phase when the flow is westerly for each group (Fig. 6d, e, and a), and afterwards, they manifest the two peaks in annual variation. The forcing tends to weaken once the flow becomes strongly easterly. The temporal variations in the indirect estimate (Fig. 6d) suggest that the variability in

wave forcing is the main driver of the observed (semi)annual behaviors of phase progression and stagnation at 30 hPa shown in Fig. 6a and b; this is ultimately confirmed by the agreement of the explicitly calculated EP flux divergence (Fig. 6e). However, it should be noted that the magnitudes of the direct estimate are roughly half those of the indirect estimate (see the discussion in Sect. 6).

Qualitatively similar results are obtained at other altitudes as well. Figures 7 and 8 present the same analysis as Fig. 6 except

at 20 hPa and 50 hPa, respectively. Common to all altitudes, the westerly-to-easterly phase progression occurs during March to July and September to December but stagnates otherwise. In other words, the months in which the phase descent typically

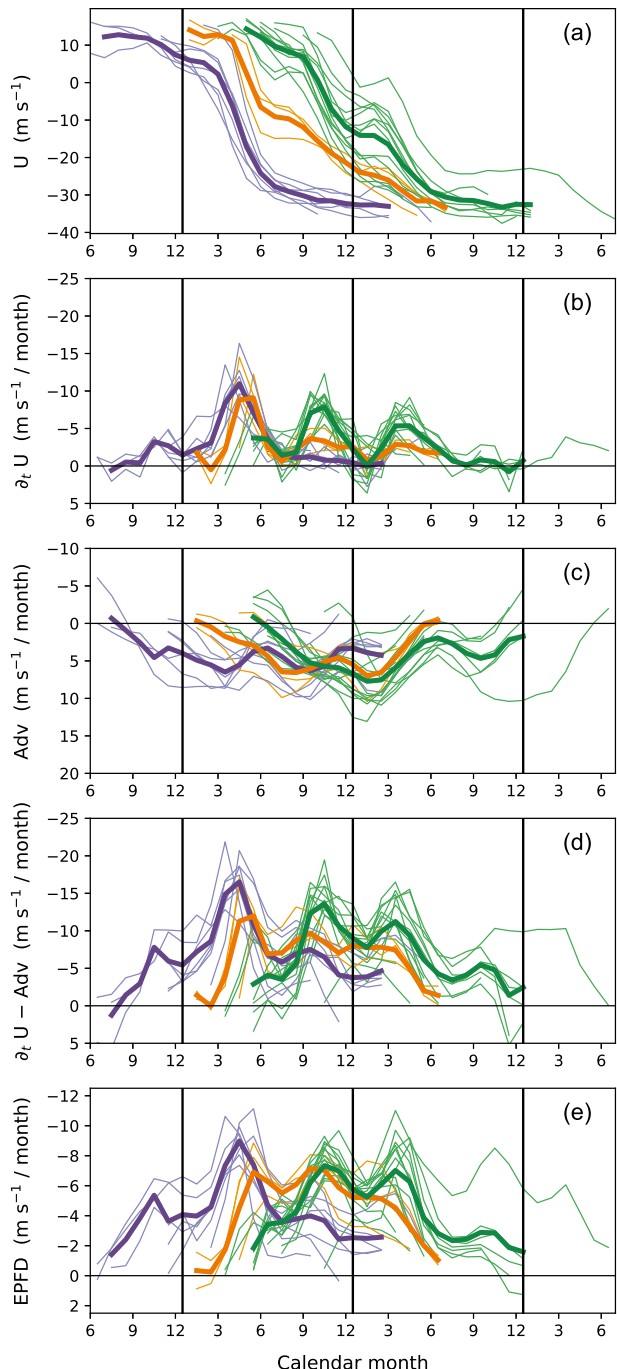

**Figure 7.** As in Fig. 6 except at 20 hPa.

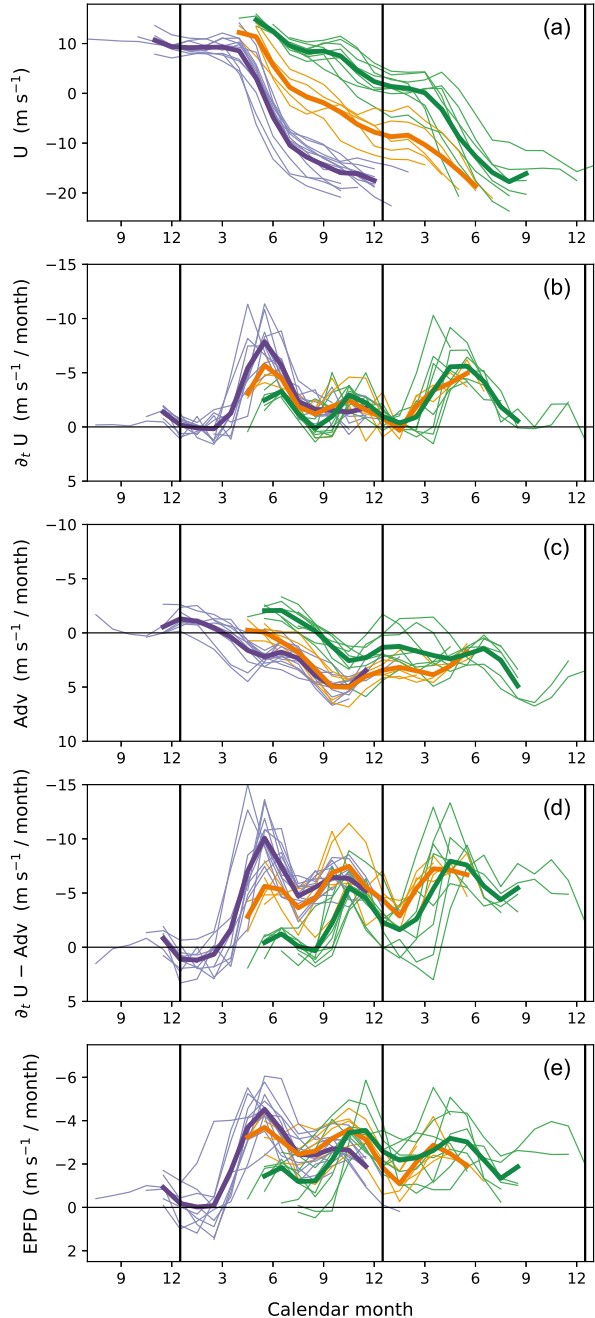

**Figure 8.** As in Fig. 6 except at 50 hPa. For better visibility, the time series are shown only after the wind at 30 hPa has transitioned to easterly.

stalls do not change with altitude (see also Fig. 5, black lines). This behavior is largely attributed to the variations in wave forcing, as the months of maximum and minimum forcing remain consistent across altitudes (panels e of Figs. 6–8). Note that these months also align with the seasonal variation in flux entering the stratosphere ($F_{84}$ shown in Fig. 5, blue dashed line), suggesting that the flux variation is responsible for them. (Accounting for more details in the evolution of wave forcing would require considering the phase-speed spectrum of flux along with the QBO shear; see Eq. (2).)

In contrast, the advection term (panels c of Figs. 6–8) does not show consistent seasonal variations between altitudes. This may result from the strong dependence of both local circulation ($\overline{v}^*, \overline{w}^*$) and zonal-wind shear on the QBO phase. Nonetheless, the advection term also seems to have a seasonal tendency at lower altitudes (e.g., Fig. 8c for 50 hPa), with relatively larger values during the latter half of the year. Across panels b of Figs. 6–8, the semiannual variability in wind tendencies decreases with decreasing altitude, accompanied by slower phase progression during September to December than during March to July. This suppressed phase progression in September to December at lower altitudes may be influenced by the seasonal tendency of the advection term. Another effect of the circulation on the annual variations in phase progression speed is discussed in the following section.

The coherent variation of the direct wave-forcing estimate (as resolved in the dataset) with the observed wind changes is striking, as this has never been revealed by long-term observational data or reanalyses (cf. for an 11-year satellite-based estimate of wave forcing, see Ern et al., 2014, Fig. 3c). Existing reanalyses, except the two most recent products (ERA5 and JRA-3Q (Kosaka et al., 2024)), have generally represented little wave forcing with indistinct monthly variations throughout the westerly-to-easterly transition phase of the QBO (see Appendix B for the results using additional reanalysis datasets including JRA-3Q).

## 5 Seasonal modulation of wave activity

The monthly variations in wave momentum flux and forcing examined in Sect. 4 are explained by the seasonal cycle of tropical convective activity and that of the wind in the tropopause layer, which both modulate the wave flux into the stratosphere. Figure 9a presents the climatological annual cycle of the upward flux of westward momentum due to the small-scale waves at 200 hPa (∼12 km) altitude. In the upper troposphere, the spatio-temporal variation of the wave-momentum flux generally follows the seasonal cycle of precipitation (white contours in Fig. 9a), which is a proxy for convective activity as the source of waves.

However, the monthly variation of the flux is significantly altered in the tropopause layer, as shown at 125 hPa (∼15 km, Fig. 9b). The 125 hPa flux exhibits two peaks, April–May and October–November, while the overall flux magnitude is reduced compared to that at 200 hPa. It is because of the dissipation of wave-momentum flux by the flow in the 200–125 hPa layer, which occurs more severely under the easterly condition for the flux of westward momentum due to gravity waves propagating westward. The flow in this layer (black contours in Fig. 9b) is easterly in the summer hemisphere. This condition, along with the weak convective activity in the winter hemisphere (Fig. 9a), results in relatively small fluxes during both solstices, with maxima occurring in April–May and October–November. A qualitatively similar monthly variation of the flux was previously

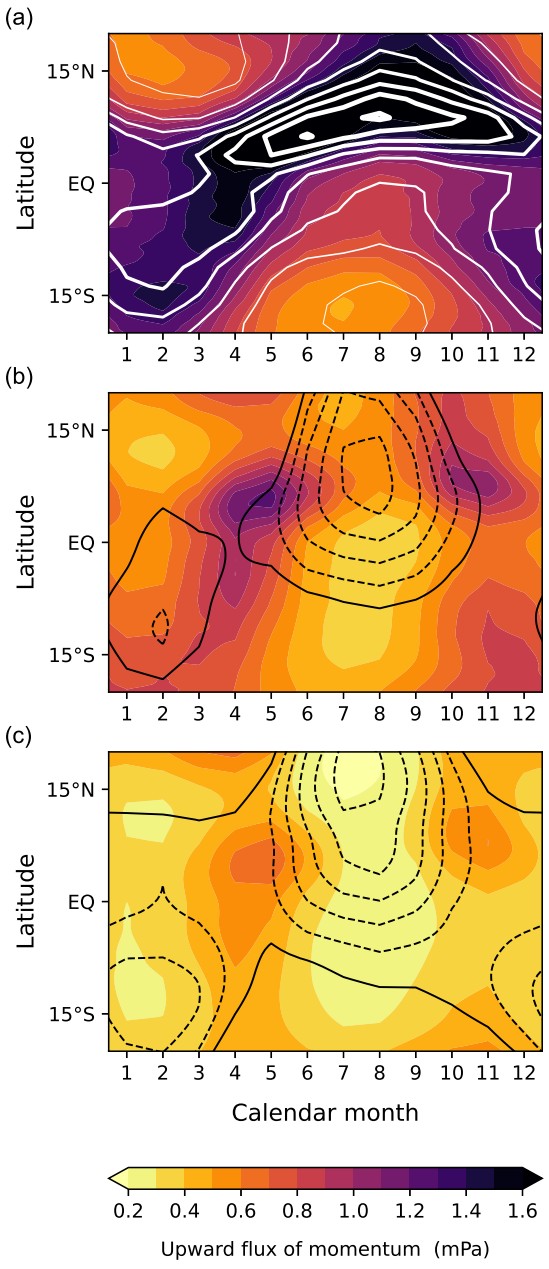

**Figure 9.** Climatological-mean annual cycle of upward flux of westward momentum due to waves with zonal wavelengths less than 2000 km (shading) at (a) 200 hPa, (b) 125 hPa, and (c) 84 hPa. The GPCP monthly precipitation rate is superimposed in (a) (white contours at intervals of $1 \, \text{mm} \, \text{d}^{-1}$). In (b) and (c), the vertical minima of zonal wind in the 125–200 hPa layer and the 84–200 hPa layer, respectively, are superimposed (black solid contours for zero winds; black dashed contours for easterlies at intervals of $3 \, \text{m} \, \text{s}^{-1}$).

obtained by Kim and Chun (2015a, Fig. 6) using a parameterization of convective gravity waves in a climate model, which
supports the role of convection and tropopause-layer wind in the flux variation.

The flux at 84 hPa (Fig. 9c) shows a similar variation to that at 125 hPa with the same months of maxima, while being further
dissipated in between. The pattern of seasonal variation does not change significantly with latitude over 15° N–15° S. For the
flux averaged over this latitude band (i.e., $F_{84}$ in the climatological mean shown in Fig. 5, blue dashed line), the maxima are
approximately 0.52 mPa in April–May and 0.45 mPa in October–November, while the minima are about 0.30 mPa in January
and 0.24 mPa in August. As noted in the regression analysis in Sect. 4.1, a large portion of the fluctuations in the descent speed
of easterly phase is explained by this flux variation, even when only its climatological seasonal cycle is considered (Fig. 3b,
grey dotted line). Notably, while the above-mentioned flux variation (0.24–0.52 mPa) corresponds to ±37% of its annual
mean, it leads to an even larger variation in the phase descent speed (about ±50% at all altitudes, Fig. 5, black lines) because
upwelling reduces the descent speed on average (see Eq. (2)). This upwelling effect in amplifying the response of descent
speed to the flux variations occurs regardless of the seasonal variability in upwelling. This is in addition to the direct effect of
upwelling variability on the descent speed at low altitudes ($p \gtrsim 40$ hPa), discussed in Sect. 4.1.

In contrast to this westward-momentum flux, the eastward-momentum flux due to small-scale waves exhibits a smaller
annual variation at 84 hPa (ranging from 0.42 mPa to 0.52 mPa) when averaged latitudinally, although it has strong seasonality
in its latitudinal distribution (not shown). Even when larger-scale waves, such as Kelvin waves, are considered collectively, the
annual variation in total eastward-momentum flux (not shown) remains within ±15% of its annual mean (∼1.25 mPa). The
relatively constant speed of the westerly-phase descent within each QBO cycle, discussed in Sect. 3 (Fig. 1a), may be attributed
to the smaller annual variation in eastward-momentum flux. Furthermore, where the westerly phase descends, the upwelling
velocity is minimal due to the QBO-induced local circulation and therefore does not significantly affect the phase-descent
speed or its variation, unlike in the opposite phase. However, a more detailed investigation may be necessary to fully explain
this phase, as the dissipation mechanism of Kelvin waves differs from that of small-scale gravity waves (e.g., Ern et al., 2009;
Krismer and Giorgetta, 2014). In addition, the QBO-westerly amplitude is relatively small (∼15 m s$^{-1}$), implying that only a
part of the wave spectrum contributes to the westerly-phase descent.

## 6  Summary and discussions

We revealed that the QBO period fluctuates primarily due to the temporal variation in the amount of westward momentum
carried by small-scale waves (gravity waves). The speed of the easterly-phase descent, the key factor controlling the QBO
period, was highly correlated with the gravity-wave momentum flux estimated in the tropical tropopause layer. Furthermore,
the stratospheric momentum forcing exerted by the waves demonstrated coherent variations with the observed phase evolutions
of the QBO. The annual variation of the wave-momentum flux was explained by the seasonal cycles of tropical convection
and tropopause-layer wind. Other sources of variation, such as the El Niño–Southern Oscillation (ENSO) (Taguchi, 2010)
and volcanic eruptions (DallaSanta et al., 2021), may also be involved on longer time scales. However, it is notable that the
annual variation already accounts for the typical behaviors of phase progression and stagnation in observed QBO cycles. The

contribution of variability in stratospheric upwelling to the fluctuation of the QBO descent speed was found to be relatively minor above ∼40 hPa altitude but became comparable to the wave contribution below that level.

Although the primary target for atmospheric reanalysis products may have been to resolve synoptic-scale weather systems and embedded larger-scale circulations, their resolutions are continuously improving beyond these scales. Our study demonstrates that waves with horizontal wavelengths less than ∼2000 km play a crucial role in QBO period fluctuations. It is encouraging that the reanalysis used here represents the interaction between these small-scale waves, convection, and the mean-flow oscillation with realistic variations, implying more opportunities for studies on multi-scale atmospheric phenomena beyond the synoptic scale. However, we also note that scales smaller than a few hundred kilometers are still largely truncated in current-generation reanalyses, including the one used here. This limitation may have led to an underestimation of the wave flux and forcing presented in our results (cf. Fig. 6d and e), suggesting that the actual impact of small-scale waves could be even more significant than our analysis indicated. In addition, although our analysis captured the temporal variations in small-scale wave activity that are highly correlated with the QBO phase speed, this does not necessarily imply that the detailed representation of convective gravity waves in the reanalysis is fully realistic, particularly when examined for individual events of convection and waves. Addressing these limitations, a comprehensive understanding of the processes, including full variability of the waves, remains essential for gaining deeper insight into the QBO.

Beyond the advancement in our understanding of QBO dynamics, these findings have important implications for both seasonal predictions and long-term climate projections. They suggest that prediction models should accurately represent the coupled variability of convection, small-scale waves, and the mean flow to improve seasonal forecasts of the QBO and related atmospheric teleconnections. This precise representation is also crucial for future climate projections of the QBO, which have shown a large spread between climate models (Butchart et al., 2020; Richter et al., 2022). By better capturing these multi-scale interactions, models may provide more reliable projections of future QBO behavior and its impacts on global climate.

*Code and data availability*. The ERA5 reanalysis data are publicly available at https://doi.org/10.24381/cds.143582cf. The GPCP monthly precipitation data can be obtained from the NOAA PSL website (https://psl.noaa.gov). The reanalysis data used in Appendix B are publicly available: ERA-Interim at https://doi.org/10.5065/D6CR5RD9; MERRA2 at https://doi.org/10.5067/QBZ6MG944HW0; JRA-3Q at https://doi.org/10.20783/DIAS.645. All analysis scripts and the resulting data are available from the corresponding author upon request.

## Appendix A: Procedure for QBO-cycle categorization

The categorization of cycles with respect to the pattern of westerly-to-easterly phase evolutions in Figs. 6–8 was motivated by their distinction into three groups as described in Sect. 4.2. For objective categorization, we used two criteria: the calendar month when the wind substantially weakened to less than $U_1$, and how the wind changed over the subsequent five months ($U_2$). With $U_1 = 0$ (except at 50 hPa where it was set to 5 m s$^{-1}$), the QBO cycles in which the wind changed below $U_1$ in January to July were categorized into the first or second groups (purple and green lines, respectively, in Figs. 6–8); the other cycles

were into the third group (orange lines). The first group was separated from the second by fast phase evolution with $U_2 < -20$, $-15$, and $-8 \, \mathrm{m\,s^{-1}}$ at 20, 30, and 50 hPa, respectively. The altitude dependence of $U_1$ and $U_2$ values was due to the vertical change in QBO amplitude.

At 30 hPa (Fig. 6), some cycles in the first and third groups (purple and orange lines, respectively) appear to exhibit a similar wind phase of $10 \, \mathrm{m\,s^{-1}}$ around March, just before undergoing distinct phase evolutions. The distinct evolutions from the similar wind phase are attributed to the larger vertical shear in the first group compared to the third, which results from differences in QBO phases at higher altitudes between the groups (not shown). The larger shear induces stronger wave forcing (Fig. 6d and e) as explained in Sect. 4.1, leading to earlier phase changes in the first group.

## Appendix B: Results from other reanalysis datasets

Three additional reanalysis products were used for comparison: ECMWF Interim Reanalysis (ERA-Interim, Dee et al., 2011), Modern-Era Retrospective analysis for Research and Applications Version 2 (MERRA-2, Gelaro et al., 2017), and Japanese Reanalysis for Three Quarters of a Century (JRA-3Q, Kosaka et al., 2024). ERA-Interim has a horizontal resolution of $\sim$80 km, spanning from 1979. MERRA-2 offers a resolution of $\sim$50 km, starting in 1980. JRA-3Q has a resolution of $\sim$40 km, from September 1947; however, we excluded data prior to 1963 due to its unrealistic representation of the QBO winds during that period.

Figures B1 and B2 show the evolutions of 30 hPa winds, tendencies, and momentum forcing during the westerly-to-easterly transition phase, as in Fig. 6, but using ERA-Interim and MERRA-2, respectively, during the period of 1980–2015. All the terms, except the EP flux divergence, exhibit qualitatively similar results with those using ERA5 (Fig. 6). However, the EP flux divergence obtained from these datasets is generally much smaller than that using ERA5, with indistinct temporal variations.

Figure B3 presents the results obtained using JRA-3Q for the period 1963–2015. The EP flux divergence exhibits notable variability, with two annual peaks that are qualitatively similar to those discussed in ERA5 (Fig. 6) but have smaller magnitudes. Overall, the magnitude of the EP flux divergence is roughly one-third of the indirect estimate of wave forcing in JRA-3Q (Fig. B3d and e).

*Author contributions.* (This paper is single-authored.)

*Competing interests.* The author declares no competing interests.

*Acknowledgements.* This research was supported by Basic Science Research Program through the National Research Foundation of Korea (NRF) funded by the Ministry of Education (RS-2023-00244727), and also supported by the NRF grant funded by the Ministry of Science and ICT (RS-2024-00342219).

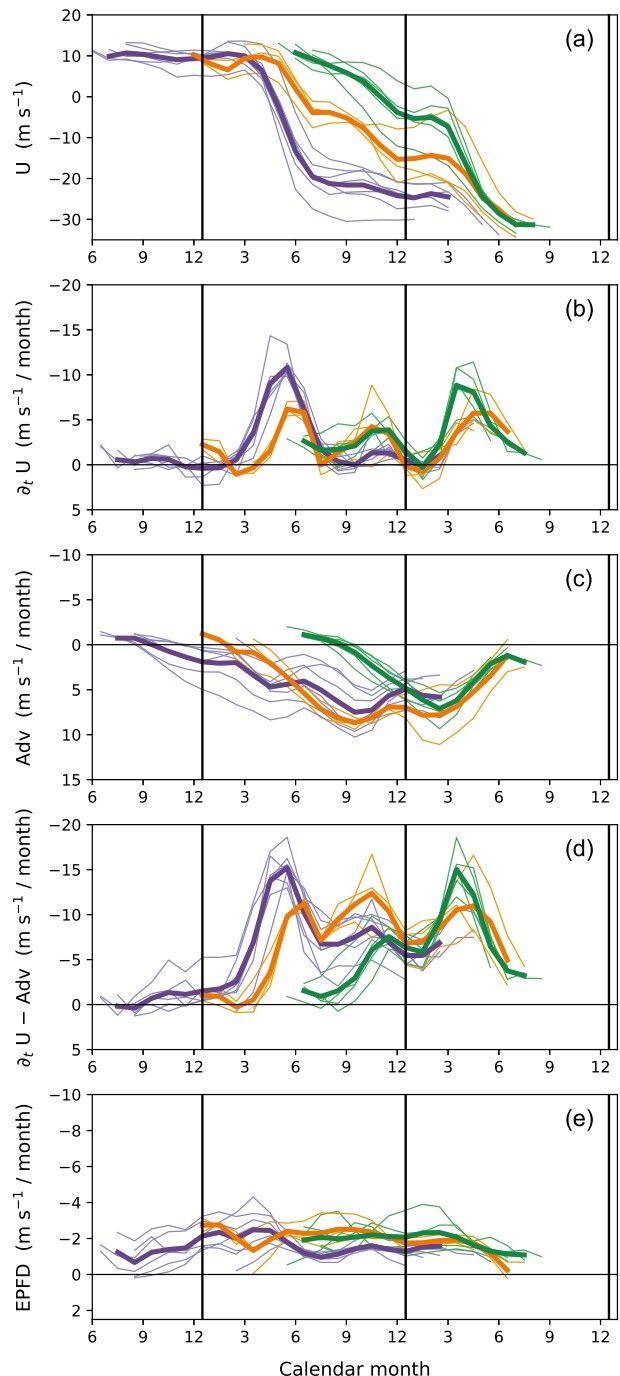

**Figure B1.** As in Fig. 6 except using ERA-Interim for the period of 1980–2015.

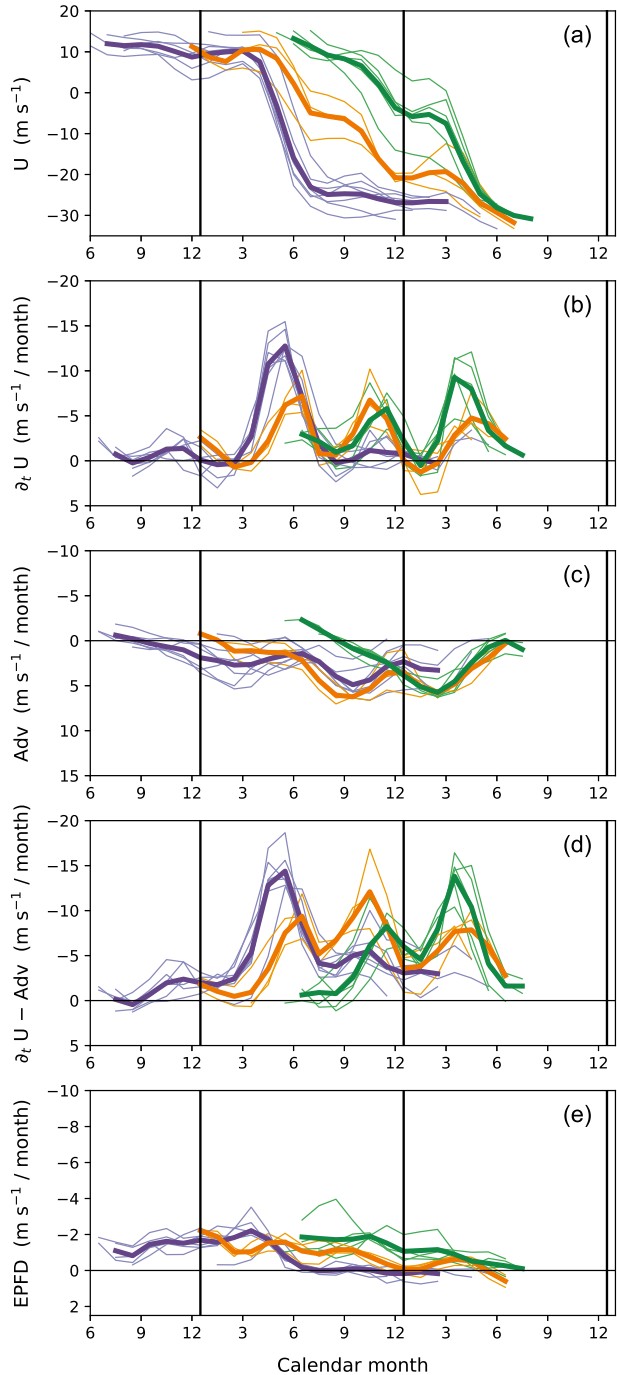

**Figure B2.** As in Fig. 6 except using MERRA-2 for the period of 1980–2015.

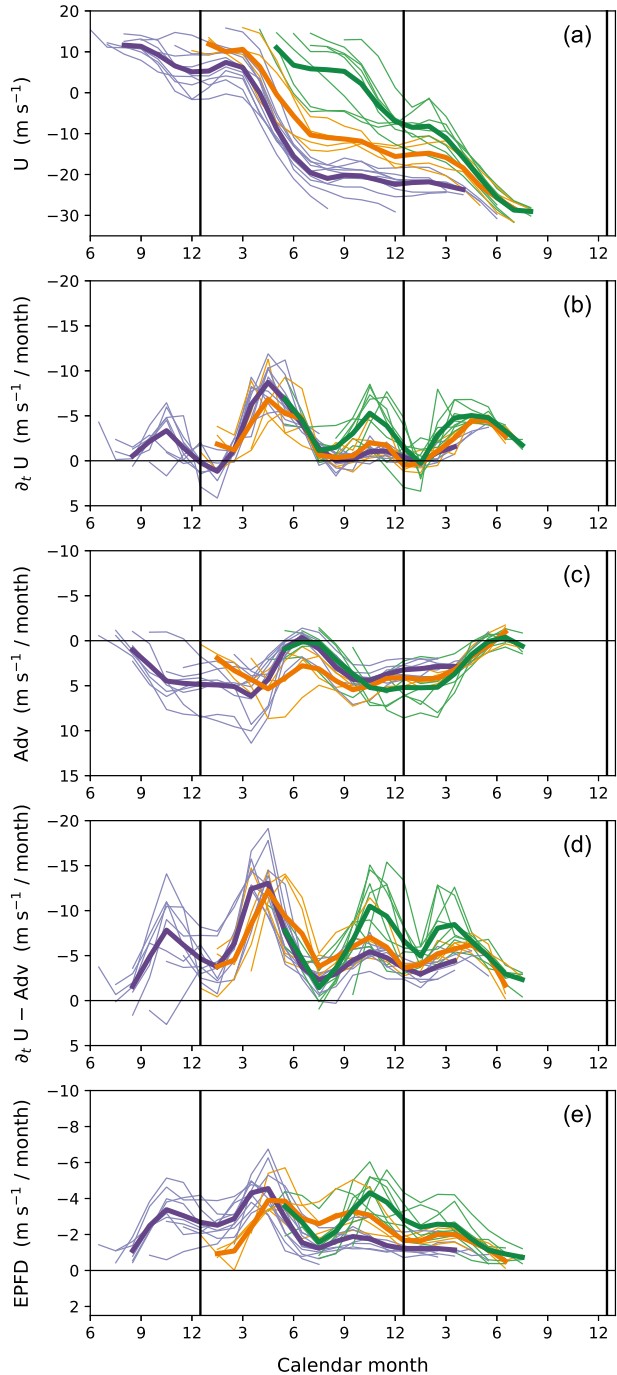

**Figure B3.** As in Fig. 6 except using JRA-3Q for the period of 1963–2015.

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
