# Peer review of "Explaining the period fluctuation of the quasi-biennial oscillation"

_EGUsphere, 2024_

## Author Comment (AC2)

**Responses to Referee #1's Comments**

The paper "Explaining the period fluctuation of the quasi-biennial oscillation" by Kim is an excellent work that addresses the long standing issue of explaining the variations in the length of the QBO period. Explaining variations of the QBO period are of great interest given the relevance of the QBO as one of the major modes of atmospheric interannual variability that has effects not only in the tropics, but also on the surface weather and climate in the extratropics. Several authors tried to explain variations of the QBO period by correlation with the solar cycle. Analysis of longer data sets, however, were less conclusive. The suggestion in this work, relating QBO period changes to variations in the forcing by small scale (gravity) waves, is therefore an interesting new explanation. The author presents a convincing chain of arguments for this mechanism.

The paper is very well written and the figures are adequate and of good quality.
The paper is therefore recommended for publication in ACP after minor revisions.
Further, the paper is recommended to be highlighted in ACP.

My main comment is that some more discussion should be added to further strengthen the paper and to provide a somewhat broader view.

Detailed comments are given below.

>> I appreciate the reviewer's comments and assessment very much. The comments help improve the manuscript. My responses to each comment are given below.

**[ Minor comments ]**

(1) In the introduction:
It should be mentioned that several papers write about the QBO period depending on the 11-year solar cycle (e.g., Salby and Callaghan, J. Climate, 2000), while analyses using longer data sets are less conclusive (e.g., Fischer and Tung, JGR, 2008; Kren et al., ACP, 2014).
Variations of wave mean flow interaction as an alternative mechanism being responsible for QBO period variations is therefore quite convincing.

>> Thank you for pointing out this aspect which I omitted in the original manuscript. This has been added in the revised manuscript [L48–51].

Do you think that wave momentum fluxes near the tropopause could be affected by the solar cycle and thereby contribute to QBO period variations related to the 11-year solar cycle?

>> Addressing this question would require a very thorough analysis, as the activity of convective wave sources is influenced by numerous meteorological factors, with the 11-year solar cycle being a relatively minor one. Additionally, it remains uncertain whether observational or reanalysis data can capture solar-cycle related variations in wave momentum fluxes. Without such an analysis, I am unable to provide a definitive opinion on this matter.

Salby, M. and Callaghan, P.: Connection between the Solar Cycle and the QBO: The Missing Link, J. Climate, 13, 2652-2662, doi:10.1175/1520-0442(1999)012<2652:CBTSCA>2.0.CO;2, 2000.

Fischer, P. and Tung, K. K.: A reexamination of the QBO period modulation by the solar cycle, J. Geophys. Res., 113, D07114, doi:10.1029/2007JD008983, 2008.

Kren, A. C., Marsh, D. R., Smith, A. K., and Pilewskie, P.: Examining the stratospheric response to the solar cycle in a coupled WACCM simulation with an internally generated QBO, Atmos. Chem. Phys., 14, 4843–4856, https://doi.org/10.5194/acp-14-4843-2014, 2014.

(2) l.54, 55: Did you use ERA5 model level data, or pressure level data provided by ECMWF?

>> The model-level data are used for the QBO wind profiles in order to obtain the descent rates at as fine vertical resolution as possible. For all the other variables presented (wave momentum fluxes, momentum forcing terms, and upwelling velocities), the pressure-level data are used. This information has been added in the revised manuscript [L61–64].

(3) l.67-70:

Please explain in more detail how wave momentum fluxes and in particular phase speeds are determined. For phase speeds the wave frequency is needed.

In l.68 it is mentioned that 2D FFT was applied for this. Usually, this requires windowing in the time domain. How was this performed? How many days were combined to perform the 2D FFT?

>> The details regarding this have been added in the revised manuscript [L73–74].

(4) l.120: It could be mentioned that some observational evidence for critical level forcing as the main main mechanism for the gravity wave forcing of the QBO is seen in the gravity wave spectra shown in Ern et al., JGR, 2014.

>> Thank you for the advice. This has been added in the revised manuscript [L132–133].

(5) After l.212:

You should give some reasoning why the descent of the QBO westerly phase is much more continuous than the descent of the QBO easterly phase.

Do you think that this is a combination of:
(a) a weaker seasonality in the sources and low-altitude filtering of the small scale waves, and
(b) large scale Kelvin waves contribute significantly to the downward propagation of the QBO westerly phase (Ern and Preusse, GRL, 2009; Kim and Chun, ACP, 2015b). Possibly, large scale Kelvin waves may have a different seasonality than the small scale waves, and the dissipation mechanism of large scale Kelvin waves is mainly radiative damping, and not critical level filtering as for the small scale waves (Ern et al., ACP, 2009; Krismer and Giorgetta, JAS, 2014).

Ern, M. and Preusse, P.: Quantification of the contribution of equatorial Kelvin waves to the QBO wind reversal in the stratosphere, Geophys. Res. Lett., 36, L21801, https://doi.org/10.1029/2009GL040493, 2009.

Ern, M., Cho, H.-K., Preusse, P., and Eckermann, S. D.: Properties of the average distribution of equatorial Kelvin waves investigated with the GROGRAT ray tracer, Atmos. Chem. Phys., 9, 7973-7995, https://doi.org/10.5194/acp-9-7973-2009, 2009.

Krismer, T. R. and Giorgetta, M.: Wave forcing of the Quasi-Biennial Oscillation in the Max Planck Institute
Earth System Model, J. Atmos. Sci., 71, 1985-2006, https://doi.org/10.1175/JAS-D-13-0310.1, 2014.

>> A new paragraph discussing this issue has been added in the revised manuscript [L242–252], along with modifications to an existing paragraph [L232–241], to provide insights on the westerly-phase descent. As commented in (a) by the reviewer, we found a weaker seasonality in the eastward momentum flux, compared to the westward momentum flux. Another reason for the continuous westerly-phase descent is overall weak local upwelling (w*) where the westerly phase descents, which therefore does not amplify the effect of the flux variation on the descent speed variation (unlike for the easterly-phase descent, as described in L238–241; 247–249). The difference in the dissipation mechanism between Kelvin and gravity waves, pointed out in (b) by the reviewer, has also been included in this paragraph.

(6) l.189/190: This is not entirely true:

It has been pointed out before by Ern et al. (2014) that during easterly shear gravity wave forcing in satellite observations and in estimates from reanalysis occurs in a series of bursts in accordance with the stepwise descent of the easterly QBO phase, while gravity wave forcing acts more continuously during westerly shear.

>> As this sentence refers to direct estimates of wave forcing from reanalyses or observations, not the indirect estimates (or 'missing drag' in the terminology of Ern et al. 2014), the most relevant result in the study of Ern et al. may be their Fig. 3c and 3d (SABER and HIRDLS estimates, respectively). I have revised the text [L211–213], citing their Fig. 3c to encourage readers to make a comparison.

(7) l.227-232: Some discussion about how realistic ERA5 resolved small scale waves are, and whether this matters, should be added:

It was shown by Preusse et. al. (2014) that convective gravity waves in the ECMWF model are not very realistic and could not be traced back to potential sources. Similarly, Okui et al. (2023) showed that in another gravity wave permitting model (JAGUAR) the agreement between model and AIRS observations in convectively dominated regions is poor.

Therefore it should be noted that for the mechanism steering the QBO period it is likely not required that the representation of convective gravity waves in ERA5 is overly realistic, as long as the spectrum of gravity waves contains the range of phase speeds interacting with the QBO. This is also confirmed by Fig.5d showing the zonal wind tendency with the contribution of advection subtracted.

Preusse, P., Ern, M., Bechtold, P., Eckermann, S. D., Kalisch, S., Trinh, Q. T., and Riese, M.: Characteristics of gravity waves resolved by ECMWF, Atmos. Chem. Phys., 14, 10483-10508, doi:10.5194/acp-14-10483-2014, 2014.
Okui, H., Wright, C. J., Hindley, N. P., Lear, E. J., & Sato, K. (2023). A comparison of stratospheric gravity waves in a high-resolution general circulation model with 3-D satellite observations. Journal of Geophysical Research: Atmospheres, 128, e2023JD038795. https://doi.org/10.1029/2023JD038795.

>> Following the reviewer's suggestion, the discussion on this issue has been added in the revised manuscript [L272–276].

(8) l.239, 240:

Please add data availability statements for ERA-Interim and MERRA2 that are used in Appendix B. Further, a statement for JRA-3Q should be added if JRA-3Q results will be provided in the revised manuscript.

>> They have been added in *Data availability* in the revised manuscript.

**[ Technical comments ]**

l.132: in independece variables -> in additional independent variables

>> This sentence has been removed.

**Responses to Referee #2's Comments**

The paper *Explaining the period fluctuation of the quasi-biennial oscillation* is an interesting work investigating the connection between variations in the length of the QBO period and variation of small-scale waves.

The paper is well-written and presents a compelling argument that variations in small-scale waves are the dominate drivers of the variation in QBO descent rates. After minor revisions towards justifying choices made in the analysis, the paper is recommended for publication in ACP.

>> I appreciate the reviewer's comments very much, which help improve the manuscript. My responses to each comment are given below.

**[ Specific comments ]**

- Hampson and Haynes 2004 posits the SAO to be a possible of the period fluctuation of the QBO (as well as upwelling and wave forcing). Can this analysis also investigate this possibility?

>> In the perspective of momentum forcing, we demonstrate that wave forcing is the primary factor for the observed period fluctuation, especially at high altitudes (Fig. 6). Therefore, the downward effect of the SAO would be reflected through the zonal-wind shear to which the wave forcing is proportional. However, temporal variations in the shear due to the SAO at 5–10 hPa (the uppermost layer of the QBO) are not in phase with the wave forcing. For instance, the SAO shear at this layer reaches its negative maximum in February whereas the maximum westward wave forcing occurs in April–May (Fig. 6e). Similarly, the second peak of the negative SAO shear (August) does not coincide with that of the wave forcing (October–November). Therefore, we think the downward effect of the SAO on the period fluctuation may be minor, although a more thorough investigation is needed to draw a definitive conclusion.

- Figure 2: To further emphasize the argument, it would be nice to see the W-descent period plotted here as as well, even if varies little.

>> Following the reviewer's suggestion, it has been added into Fig. 2, with modifications to the text in the revised manuscript [L112–123] to discuss this additional result.

- Why use monthly rather than weekly means? In (page 5, line 107) the limitations of lower frequency sampling is noted.

>> Ultimately, we relate the descent speed to the 70-hPa wave flux, using monthly data. If this analysis were conducted on a weekly basis, it would be necessary to account for the time lag between the two time series, as it can take up to a few days for waves to travel from 70 hPa to ~10 hPa. However, the travel time varies largely, depending on wave scales (e.g., horizontal wavelength) and mean-flow states along the propagation paths, which would complicate the regression analysis. By using monthly means, we can simply disregard time lags of up to a few days.

- 4.1, page 6, 120-133: Please justify choice of F70 (rather than say, F50). Do these show similar annual evolution? Similarly, comment on column averaged w* rather than w* at several levels. In particular, I would naively expect that height at which w* is measured/regressed would matter for descent rates.

>> In order to explain the dependent variable using independent variables via a regression analysis, the independent variables should not contain variability arising from the dependent variable (otherwise, the causality may not be clear between the variables). In our regression, this means that the wave flux and upwelling velocity should not contain QBO-related variations so that the descent rate (i.e., QBO phase speed) is explained by the two independent quantities. The 70-hPa altitude is chosen as the highest level with a little impact of the QBO on the wave flux (and also it is conventionally the bottom of the tropical stratosphere, as required in Eq. (2); refer to the definition of $F(c)$). In the same reason, w* has been averaged to reduce the effect of the QBO-induced local circulation on it. This is also in line with the argument of the previous studies (introduced in Section 1) that the seasonal cycle of the "stratospheric upwelling" that is driven by extratropical planetary-wave forcing could modulate the QBO descent speed.

Yes, a similar annual evolution is maintained in F50 (see Fig. R1). Regarding the use of local w*, interestingly, it did not yield an overall better regression score compared to W (see Fig. R2; cf. Fig. 3b). The text has been revised to better clarify the rationale of using F70 and W [L144–146].

[Figure]

Fig. R1. As in Fig. 8 except at (a) 70 hPa and (b) 50 hPa, with a different color scale.

[Figure]

Fig. R2. As in Fig. 3b except using w* at each altitude.

- page 6, 138-139: Please clarify the phrase "the selected time series were standardized."

>> This has been clarified in the revised manuscript [L153].

- Figure 5 (and similar): Could the y-axes be adjusted to be consistent throughout the figure? I think this would aid in comparison between panels 5d and 5e greatly.

>> The y-axes for the tendency variables share the same scale interval except the one in panel (e) (which uses the 50% smaller scale interval). This has been clarified in the figure caption in the revised manuscript: "*Note that the y-axes in (b)–(d) use the same scale interval, and the scale in (e) is half that.*" I believe this is the best way to compare the magnitudes across the variables without wasting spaces.

- Figure 5 (and similar): Can comment on why the wave forcing estimates (panels d and e) appear to be "out of phase" with one another in the first half of the year? If the variations in wave-forcing are primarily driven by the seasonal cycle, I would expect this not to be the case. (Something like ENSO perhaps?).

>> I could not observe an out-of-phase relationship between the two estimates (panels d and e). Instead, my response assumes that the reviewer referred to the substantial spread among cycles within each group in a given calendar month.

Wave forcing depends not only on variations in wave flux but also significantly on the wind and its vertical shear associated with the QBO phase, given the critical-level filtering process (refer to Section 4.1, first paragraph). Even with similar wave fluxes at the tropopause, small differences in the detailed sheared-wind structures due to the QBO can lead to substantial differences in wave forcing. Thus, the notable spread within each group (thin lines of the same color) for a given calendar month does not undermine the importance of the seasonal cycle in wave flux variability for QBO phase evolutions. On the contrary, we emphasize its critical role, as shown in the prediction using monthly climatologies (Fig. 3b, grey dotted lines). Even so, variability beyond the seasonal cycle (such as ENSO) may also partly contribute to these variations (as mentioned in L259–260 in the revised manuscript), and this warrants further investigation in future studies.

- Appendix A: I don't understand why we get to separate the purple and orange lines, as they seem to "start" at the same place.

>> It is because the wind shear is in different states between the two groups when they begin to diverge (e.g., around March, Fig. 5a), although the wind itself is similar at that timing. The wave forcing depends on both the wind and shear, as described in Section 4.1 (the first paragraph). The larger shear in the purple group (due to faster phase evolutions at higher altitudes, although not identifiable in Fig. 5) results in larger wave forcing around March (Fig. 5e), leading to earlier phase changes at the target altitude, compared to the orange group. This explanation has been included in Appendix A in the revised manuscript [L296–300].

---

## Referee Report (RR1)

**Review of *Explaining the period fluctuation of the quasi-biennial oscillation* by Young-Ha Kim**

February 12, 2025

This paper offers a new explanation for the source of period fluctuations of the QBO. Such fluctuations are hypothesized to arise from seasonality in tropical convection, which launches small-scale gravity waves that are then filtered through the seasonally-evolving winds of the tropical tropopause layer to lead to a seasonally-varying gravity wave stress at the base of the QBO, which then contributes to a seasonally-varying descent rate. This contribution to the seasonally-varying descent rate is hypothesized to explain seasonality in the total descent rate, which in turn drives cycle-to-cycle variability of the QBO period. This cycle-to-cycle variability in the QBO period is noted to have implications for QBO understanding and predictability.

There are several steps to the argumentation in the paper, all of which must be true in order to justify the ultimate conclusions. Some of the steps are already convincing, but some would benefit from minor revisions in order to be convincing. Here are the key steps:

1. Variability in the QBO period is dominated by variability by in the descent rates of easterly shear zones (well-justified)

2. Variability in the descent rates of easterly shear zones is driven by variability in small-scale gravity wave stress propagating up through the tropopause (would benefit from minor revisions)

3. Seasonality in small-scale gravity wave stress propagating up through the tropopause results when waves that are launched by tropical convection are filtered through the seasonally-varying winds in the TTL (well-justified)

The paper should more forcefully disentangle the relative roles of seasonality in its newly-proposed mechanism (small-scale gravity wave stress) versus the commonly-accepted mechanism (residual upwelling). Towards that end, Step #2 must establish that variability in small-scale easterly gravity wave stress crossing the tropopause ($F_{70}$) is the primary driver of variability in the QBO period (e.g., as stated on Lines 254-255 of the Conclusions). The primary importance of $F_{70}$ must be established counter to the commonly-accepted view

that the primary driver is variability of residual upwelling ($W$). Numerous previous papers have put forward compelling evidence for the importance of $W$, including quantitative agreement between the annual cycle in $W$ and the annual variations in the QBO descent rate (e.g., Coy et al., 2020).

The paper offers two broad pieces of evidence in favor of F70. The first broad argument is based on multi-linear regression in which F70 and W are used to predict descent rate, presented in Section 4.1, Eq. 3, and Fig. 3. This argument is not fully convincing because F70 and W appear to have very similar seasonal effects on QBO descent rate: comparing Fig. 1 from Coy et al. (2020) and Fig. 8c of the submitted paper makes it clear that both F70 and W contribute to minimum QBO descent rates in NH winter and August/September, and larger descent rates otherwise. Because F70 and W are strongly anti-correlated over the seasonal cycle, their effects are probably, at least in part, statistically degenerate. This plausible degeneracy calls into question the suitability and uncertainty of the multi-linear regression in Eq. 3 and plotted in Fig 3. If this correlational analysis is intended to be included, it is minimally necessary to do the following:

- plot the uncertainties in the regression coefficients in Fig. 3b, which hopefully take into account the effects of any collinearity. If the uncertainty bars are large and overlapping, this must change the interpretation of the results.

- plot the average seasonal cycles of F70 and W so that readers can visually distinguish how distinct they are.

Alternatives to this correlational analysis are possible. For example, I expected that the paper was going to take advantage of the descent rate formalism in Equation 2 to formulate a quantitative descent rate budget, i.e.,:

$$\frac{\partial \bar{u}}{\partial t}(\frac{\partial \bar{u}}{\partial z})^{-1} = \frac{\nabla \cdot F}{\rho_0 a cos \phi}(\frac{\partial \bar{u}}{\partial z})^{-1} - \bar{w}^* + \bar{v}^* \hat{f}(\frac{\partial \bar{u}}{\partial z})^{-1} \qquad (1)$$

Using such a budget, it would be possible to make quantitative arguments about the role of EPFD (potentially decomposed by wavenumber) and upwelling in driving descent rates. This argument would be stronger than the correlations, because the magnitudes of each term matter and not just their temporal structure. If it were found that the local EPFD at a given altitude is a dominant driver of the seasonality in local descent rates, then it would have to be subsequently established that the local EPFD at a given altitude scales with the incoming small-scale gravity wave stress crossing 70 hPa, per the assumed form of EPFD inspired by Lindzen and Holton (1968). This would form a stronger quantitative link in terms of temporal structure AND magnitude rather than the paper's current argument of Section 4.1 that is only in terms of temporal structure.

The magnitudes are considered in Section 4.2, which is good, although it shows wind tendencies, so to gain insight into descent rates, the reader must attempt to divide in their head by the hypothetical time-evolving vertical shear. Showing descent rates directly could

be valuable. Also, is the EPFD plotted in Figs. 5/6/7 the total EPFD or just that from small-scale waves? Please ensure that all claims about small-scale waves are not based on any evidence from EPFD integrated across all wavenumbers as opposed to just the small scales.

Two last minor considerations:

- Line 142: "The spatial averaging applied to W aimed to reduce the effect of the QBO-induced local circulation on it (for details on the local circulation, see Plumb and Bell, 1982, Fig. 1)." Does "spatial averaging" refer to the averaging in the vertical (over 10-70 hPa) or the averaging in the horizontal (15S-15N)? In either case, the effects of the QBO-induced secondary circulation might not be particularly reduced. When averaging vertically, the secondary circulation can still project onto the domain-wide mean upwelling, because there is often only one strong shear zone in the domain (the other being stalled against the lower boundary). When averaging horizontally from 15S-15N, this only includes the tropical branch of the secondary circulation but not the extratropical branch, which is located poleward of roughly 15 degrees (e.g., Randel et al., 1999; Baldwin et al., 2001), so this will not reduce the effects of the meridional structure of the QBO-induced circulation.

- Lines 238-241 states: "Notably, while the above-mentioned flux variation (0.19-0.37 mPa) corresponds to ±32% of its annual mean, it leads to an even larger variation in the phase descent speed because upwelling reduces the descent speed on average (see Eq. (2)). This upwelling effect in amplifying the response of descent speed to the flux variations occurs regardless of the seasonal variability in upwelling." I am confused by this argument. Isn't descent rate linear in both upwelling and EPFD, per Equation 2? Perhaps the paper intended to refer to fractional variations in the phase descent speed?

**References**

Baldwin, M. P., and Coauthors, 2001: The quasi-biennial oscillation. *Reviews of Geophysics*, **39 (2)**, 179, doi:10.1029/1999RG000073.

Coy, L., P. A. Newman, S. Strahan, and S. Pawson, 2020: Seasonal Variation of the Quasi-Biennial Oscillation Descent. *Journal of Geophysical Research: Atmospheres*, **125 (18)**, e2020JD033 077, doi:10.1029/2020JD033077.

Lindzen, R. S., and J. R. Holton, 1968: A Theory of the Quasi-Biennial Oscillation. *Journal of the Atmospheric Sciences*, **25 (6)**, 1095–1107, doi:10.1175/1520-0469(1968)025⟨1095: ATOTQB⟩2.0.CO;2.

Randel, W. J., and Coauthors, 1999: Global QBO Circulation Derived from UKMO Stratospheric Analyses. *Journal of the Atmospheric Sciences*, **56 (4)**, 457–474, doi: 10.1175/1520-0469(1999)056⟨0457:GQCDFU⟩2.0.CO;2.

---

## Author Response (AR2)

**Responses to Referee #3's Comments**

This paper offers a new explanation for the source of period fluctuations of the QBO. Such fluctuations are hypothesized to arise from seasonality in tropical convection, which launches small-scale gravity waves that are then filtered through the seasonally-evolving winds of the tropical tropopause layer to lead to a seasonally-varying gravity wave stress at the base of the QBO, which then contributes to a seasonally-varying descent rate. This contribution to the seasonally-varying descent rate is hypothesized to explain seasonality in the total descent rate, which in turn drives cycle-to-cycle variability of the QBO period. This cycle-to-cycle variability in the QBO period is noted to have implications for QBO understanding and predictability.

>> I sincerely appreciate the referee's comments and assessment. The comments help improve the manuscript. During the revision, we have changed the altitude for the wave flux from 70 hPa (F70) to 84 hPa (F84) in the regression analysis, according to the Editor's comment, and updated all relevant figures (Figs. 3, 4, and 9 in the revised manuscript) and text, while the main conclusions remain unchanged. Also, following the referee's comment, we have added a new figure presenting the seasonal cycles of W and F84 (Fig. 5) and its discussion (L176–198). My responses to each comment are given below.

There are several steps to the argumentation in the paper, all of which must be true in order to justify the ultimate conclusions. Some of the steps are already convincing, but some would benefit from minor revisions in order to be convincing. Here are the key steps:

1. Variability in the QBO period is dominated by variability by in the descent rates of easterly shear zones (well-justified)
2. Variability in the descent rates of easterly shear zones is driven by variability in small-scale gravity wave stress propagating up through the tropopause (would benefit from minor revisions)
3. Seasonality in small-scale gravity wave stress propagating up through the tropopause results when waves that are launched by tropical convection are filtered through the seasonally-varying winds in the TTL (well-justified)

The paper should more forcefully disentangle the relative roles of seasonality in its newly-proposed mechanism (small-scale gravity wave stress) versus the commonly-accepted mechanism (residual upwelling). Towards that end, Step #2 must establish that variability in small-scale easterly gravity wave stress crossing the tropopause ($F_{70}$) is the primary driver of variability in the QBO period (e.g., as stated on Lines 254-255 of the Conclusions). The primary importance of $F_{70}$ must be established counter to the commonly-accepted view that the primary driver is variability of residual upwelling (W). Numerous previous papers have put forward compelling evidence for the importance of W, including quantitative agreement between the annual cycle in W and the annual variations in the QBO descent rate (e.g., Coy et al., 2020).

The paper offers two broad pieces of evidence in favor of F70. The first broad argument is based on multi-linear regression in which F70 and W are used to predict descent rate, presented in Section 4.1, Eq. 3, and Fig. 3. This argument is not fully convincing because F70 and W appear to have very similar seasonal effects on QBO descent rate: comparing Fig. 1 from Coy et al. (2020) and Fig. 8c of the submitted paper makes it clear that both F70 and W contribute to minimum QBO descent rates in NH winter and August/September, and larger descent rates otherwise. Because F70 and W are strongly anti-correlated over the seasonal cycle, their effects are probably, at least in part, statistically degenerate. This plausible

degeneracy calls into question the suitability and uncertainty of the multi-linear regression in Eq. 3 and plotted in Fig 3. If this correlational analysis is intended to be included, it is minimally necessary to do the following:

• plot the uncertainties in the regression coefficients in Fig. 3b, which hopefully take into account the effects of any collinearity. If the uncertainty bars are large and overlapping, this must change the interpretation of the results.

• plot the average seasonal cycles of F70 and W so that readers can visually distinguish how distinct they are.

>> Thank you for pointing out this issue. The responses to the two suggestions are provided below for each.

• We have confirmed that the correlation coefficient between F84 and W is very small: within −0.13 to +0.09 across different altitudes. Therefore, statistical degeneration in the regression is not expected. This information has been added in the revised manuscript [L188−190]. Additionally, we have confirmed that similar results are obtained even when the regression analysis is conducted separately with F84 and W, as seen in Fig. C1 (cf. Fig. 3b). This outcome is expected given their low correlation.

[Figure]

[Figure]

Fig. C1. Standardized regression coefficient of (left) F84 and (right) W for the descent speed at each altitude, obtained separately from single-variable linear regression.

• To further investigate relations between F84 and W (along with the descent speed) in their seasonal cycles, we have added a new figure (Fig. 5) and corresponding text at the end of Sect. 4.1 in the revised manuscript [L176−198], following the referee's comment.

In the newly added Fig. 5, W at 20−50 hPa exhibits a dominant annual cycle with only minor semiannual variation. In contrast, F84 is primarily driven by a semiannual cycle, resulting in a low correlation with W (less than ±0.13).

As noted by the referee, Coy et al. (2020, Fig. 1) showed a semiannual cycle in w*. There are a couple of potential factors that would lead to the difference between that and our result.

(1) We use the vertically and meridionally averaged w* over 15°N–15°S (defined as W), whereas Coy et al. showed 10-hPa w* over 10°N–10°S.

(2) Our regression is performed only for a specific QBO phase at each altitude (when the descending easterly phase $U_\varphi = -10$ m/s occurs at the altitude). The seasonal cycle is constructed exclusively from W values at that phase.

To examine these two factors, Fig. C2 shows the same analysis as Fig. 5 except that w* at each altitude over 10°N–10°S (following Coy et al.) is plotted instead of W, and that the 10-hPa climatological annual cycles of w* and F84 are included. (Unfortunately, regression analysis was not possible for 10 hPa due to excessively fast descents, as noted in L119.)  First, it is confirmed that the w* variation at 10 hPa in ERA5 is qualitatively consistent with Coy et al. (2020, Fig. 1b) using MERRA2, exhibiting a similar semiannual variation (although some differences exist in details, such as maximum speed of w*). (Note that an inverted axis is used for w* in Fig. C2.)

However, from 30 hPa downward, the climatological annual cycles of w* become qualitatively similar to W, showing decreased semiannual variability (compare the orange dashed lines between Figs. 5 and C2). Moreover, when the cycles are derived for the easterly-descending phase at each altitude (i.e., from the subset of w* values, as employed in the regression) (solid lines), w* exhibits similar variations to W, from 20 hPa downward. This suggests that the regression result might remain roughly similar even if the altitude-dependent w* is used instead of W. This has been confirmed in Fig. C3. However, in this case, the correlation between w* and F84 was relatively high at some altitudes (~0.3 at 30 hPa and above; ~0.1 at 40 hPa), introducing uncertainty into the results in Fig. C3.

In our study, the spatially averaged W rather than w* is used to minimize the QBO-induced factor (i.e., the QBO-induced local-circulation component in w*) on the RHS of the regression, so that the descent speed variation can be causally explained by the two *external* factors [L149–150]. In other words, if w* were used, part of its variation could potentially be a response to descent speed variation, complicating result interpretation. Beyond this conceptual reasoning, the low correlation between W and F84 (as discussed above) further supports our choice of W for the regression analysis. Additionally, using W aligns with many previous studies that have focused on the annual variation of tropical upwelling (forced by midlatitude Rossby waves), rather than its semiannual variation at ~10 hPa, to explain the descent speed variation (e.g., Kinnersley and Pawson, 1996; Hampson and Haynes, 2004; Rajendran et al., 2018).

[Figure]

Fig. C2. Same as Fig. 5 except using w* at each altitude over 10°N–10°S, instead of W, and adding the 10-hPa climatological annual cycles of w* and F84.

[Figure]

Fig. C3. Same as Fig. 3b except that w* at each altitude over 10°N–10°S is used instead of W.

Alternatives to this correlational analysis are possible. For example, I expected that the paper was going to take advantage of the descent rate formalism in Equation 2 to formulate a quantitative descent rate budget, i.e.,:

$$\frac{\partial \bar{u}}{\partial t}\left(\frac{\partial \bar{u}}{\partial z}\right)^{-1} = \frac{\nabla \cdot F}{\rho_0 a cos\phi}\left(\frac{\partial \bar{u}}{\partial z}\right)^{-1} - \bar{w}^* + \bar{v}^*\hat{f}\left(\frac{\partial \bar{u}}{\partial z}\right)^{-1} \qquad (1)$$

Using such a budget, it would be possible to make quantitative arguments about the role of EPFD (potentially decomposed by wavenumber) and upwelling in driving descent rates. This argument would be stronger than the correlations, because the magnitudes of each term matter and not just their temporal structure. If it were found that the local EPFD at a given altitude is a dominant driver of the seasonality in local descent rates, then it would have to be subsequently established that the local EPFD at a given altitude scales with the incoming small-scale gravity wave stress crossing 70 hPa, per the assumed form of EPFD inspired by Lindzen and Holton (1968). This would form a stronger quantitative link in terms of temporal structure AND magnitude rather than the paper's current argument of Section 4.1 that is only in terms of temporal structure.

The magnitudes are considered in Section 4.2, which is good, although it shows wind tendencies, so to gain insight into descent rates, the reader must attempt to divide in their head by the hypothetical time-evolving vertical shear. Showing descent rates directly could be valuable. Also, is the EPFD plotted in Figs. 5/6/7 the total EPFD or just that from small-scale waves? Please ensure that all claims about small-scale waves are not based on any evidence from EPFD integrated across all wavenumbers as opposed to just the small scales.

>> Since the correlation and the related analysis (newly added Fig. 5) address the first suggestion (see my response above), this alternative approach is not further examined. As noted above, the local w* not only exhibits variations that modulate the descent speed but may also potentially contain direct or indirect responses to the descent speed variation. Therefore, we do not extend the analysis using w* beyond the existing one (Figs. 6–8). Nonetheless, I greatly appreciate your insightful suggestion.

Regarding the EPFD, it represents the total wave forcing from waves of all scales. This has been clarified in the revised manuscript [L91; L225; and in captions of Figs. 6–8 by referring to Eq. (1)]. The use of total wave forcing (without spectral filtering) is justified, as westward QBO forcing by large-scale waves is known to be minor (e.g., Ern et al., 2014; Kim and Chun, 2015b), as stated in L144–145. Additionally, we have included a result confirming the limited contribution of large-scale waves to descent speed variation, based on an additional regression analysis (not shown) in the revised manuscript [L199–202].

Two last minor considerations:

• Line 142: "The spatial averaging applied to W aimed to reduce the effect of the QBO-induced local circulation on it (for details on the local circulation, see Plumb and Bell, 1982, Fig. 1)." Does "spatial averaging" refer to the averaging in the vertical (over 10-70 hPa) or the averaging in the horizontal (15S-15N)? In either case, the effects of the QBO-induced secondary circulation might not be particularly reduced. When averaging vertically, the secondary circulation can still project onto the domain-wide mean upwelling, because there is often only one strong shear zone in the domain (the other being stalled against the lower boundary). When averaging horizontally from 15S-15N, this only includes the tropical branch of the secondary circulation but not the extratropical branch, which is located poleward of roughly 15 degrees (e.g., Randel et al., 1999; Baldwin et al., 2001), so this will not reduce the effects of the meridional structure of the QBO-induced circulation.

>> The spatial averaging is applied in both the vertical and horizontal directions [L141–142]. While I agree that this averaging cannot fully eliminate the QBO-induced circulation especially when the vertical layer is predominantly under either easterly or westerly shear, it helps reduce its influence to some extent. For example, the differences between orange dashed and solid lines in Fig. C2 (seasonal climatology vs. seasonal cycle derived from the easterly-descending phase at a given altitude) reflect the QBO-phase dependence of w*. Fig. C2 shows that w* at the easterly-descending phases is consistently larger than the climatology (note the inverted axis). In contrast, the differences between the dashed and solid lines in Fig. 5 are much smaller, suggesting that the spatial averaging for W substantially reduces the secondary-circulation component.

In addition, using a broader latitude band to include the extratropical upwelling branch may be less appropriate, because (1) the upwelling branch shifts north and south with seasons, with its boundary located near ~15° in the winter hemisphere (depending on altitude), and (2) w* outside of ~15° is unlikely to significantly affect QBO dynamics, given the latitudinal width of the QBO.

• Lines 238-241 states: "Notably, while the above-mentioned flux variation (0.19-0.37 mPa) corresponds to ±32% of its annual mean, it leads to an even larger variation in the phase descent speed because upwelling reduces the descent speed on average (see Eq. (2)). This upwelling effect in amplifying the response of descent speed to the flux variations occurs regardless of the seasonal variability in upwelling." I am confused by this argument. Isn't descent rate linear in both upwelling and EPFD, per Equation 2? Perhaps the paper intended to refer to fractional variations in the phase descent speed?

>> Yes, this statement refers to fractional variations. This is clarified in the revised manuscript [L279] by explicitly stating the fractional value for the phase speed variation as a percentage ("about ±50%").

**References**

Baldwin, M. P., and Coauthors, 2001: The quasi-biennial oscillation. Reviews of Geophysics, 39 (2), 179, doi:10.1029/1999RG000073.

Coy, L., P. A. Newman, S. Strahan, and S. Pawson, 2020: Seasonal Variation of the Quasi-Biennial Oscillation Descent. Journal of Geophysical Research: Atmospheres, 125 (18), e2020JD033 077, doi:10.1029/2020JD033077.

Lindzen, R. S., and J. R. Holton, 1968: A Theory of the Quasi-Biennial Oscillation. Journal of the Atmospheric Sciences, 25 (6), 1095–1107, doi:10.1175/1520-0469(1968)025<1095:ATOTQB>2.0.CO;2.

Randel, W. J., and Coauthors, 1999: Global QBO Circulation Derived from UKMO Stratospheric Analyses. Journal of the Atmospheric Sciences, 56 (4), 457–474, doi:10.1175/1520-0469(1999)056<0457:GQCDFU>2.0.CO;2.

>> All the references cited in my response are included in the manuscript.

**Responses to Editor's Comments**

>> I greatly appreciate the editor's comments, which have helped strengthen the manuscript. During the revision, we have changed the altitude for the wave flux from 70 hPa to 84 hPa in the regression analysis, and updated all relevant figures (Figs. 3, 4, and 9 in the revised manuscript) and text, while the main conclusions remain unchanged. A new figure (Fig. 5) has also been added following Referee #3's comment. My responses to each comment are given below.

**Major comments:**

L144-145 - By using the QBO-independent variables (instead of flux at a higher altitude or local w ), the regressed descent speed can be causally attributed to these variables. - This is a crucial assumption of the paper and it seems not valid. For this you would have to take F and W lower down, let's say at 100hPa. Because, as your Fig. 8 clearly shows, F70 is itself a subject to filtering by the winds below 70 hPa in the UTLS, which are most likely not independent on QBO.

>> We acknowledge that F70 may be influenced by the QBO to some extent. As for an alternative choice, the 100 hPa altitude is however too close to the convective source region in the tropics, where the flux is often affected by diabatic heating from convective and anvil clouds, making it less representative of freely propagating waves. Therefore, in the revised manuscript, we have chosen the 84 hPa altitude from ERA5 model levels as a more suitable alternative and replaced F70 with flux at this altitude (F84). The relevant figures (Figs. 3, 4, and 9) have also been updated, accordingly. Importantly, our main conclusions remain unchanged.

**Minor and technical comments (chronological ordering):**

L30 not only waves from convection propagate vertically to the stratosphere.

>> This comment is valid. However, since this paragraph begins with a focus specifically on the QBO, I have chosen to restrict the discussion to convective waves here (with references only relevant to them).

L62 It is strange that for the descent speed estimation you needed native model-level dataset, but momentum forcing, which relies on vertical divergence estimation that is crucially sensitive to the vertical resolution, has been calculated on pressure levels (and hence can be subject to large errors). Please explain.

>> I agree that using native model-level dataset would provide more accurate quantities (but due primarily to the vertical truncation for flux quantities; e.g., Kim & Chun, 2015, ACP, doi:10.5194/acp-15-6577-2015). However, there are two reasons why the pressure-level based quantities are used here: (1) Most previous studies, including multi-model and multi-reanalysis studies, have used standard pressure-level data for TEM calculations. Consequently, providing TEM results based on model-level data made quantitative comparisons challenging. (2) I no longer have access to archived model-level data for reanalyses other than ERA5 (used in Appendix), and independently collecting these datasets is highly impractical.

Thus, using pressure-level data ensures compatibility with existing studies showing TEM results and consistency in this study. Nonetheless, for the descent speed, the key quantity of the study, using fine

vertical grids was crucial because it varies rapidly with height, as seen in Fig. 1a (refer to the inflection of the zero-wind line between 30 and 50 hPa from late 1997 to early 1998). The descent-speed calculation involves differentiation along the vertical trajectory, making vertical truncation a significant issue.

L69 Please define the 1-2-1 smoothing

>> We have modified it to "1-2-1 temporal smoothing" in the revised manuscript [L71]. We believe this clarifies the methodology, as the 1-2-1 smoothing method is widely recognized.

L71 Monthly fluxes have been calculated ..after decomposing the hourly data?

>> The expression has now been clarified: "after decomposing waves …" [L74]

L73-74 d→day?

>> 'd' is the correct unit notation in the EGU convention.

L75-76 - Analytical description of this conditional summing in Fourier domain would be beneficial. What about possible orographic sources?

>> Unfortunately, there is no simpler/shorter mathematical expression for the sum of positive-valued momentum fluxes with positive zonal phases (and for the negatives as well). The condition should anyway be written in a descriptive form.
    Given that waves over the tropics are generated primarily by convective sources, we have neglected the possible contribution of orographic waves to the monthly and zonally averaged flux.

L80 Please provide a detailed reasoning, why you expect the noise to dominate the waves with small phase speeds. Generally, it is assumed that the effects of noise should be symmetrical and isotropic.

>> The detailed reason has been included in the revised manuscript [L81–84].

Fig.1 - Why different averaging domains for winds and momentum fluxes? All aspects like this must be clearly justified. Is the meridional propagation of GWs one of the reasons for this?

>> Yes, the meridional propagation of GWs is the reason for that [L147].

L90 -indirect estimate- Like this you group into this indirect estimate also the parameterized tendencies, analysis increments + possible inaccuracy of your computation. This should be clearly stated here.

>> Equation (1) is the generally applied momentum equation, not specific to reanalyses (no external forcing from unresolved/parameterized processes or data assimilation is introduced on its RHS). Therefore, the EP flux in Eq. (1) accounts for the total wave forcing from waves of all scales. We have clarified this in the revised manuscript [L91–92]. The indirect estimate, therefore, represents the same quantity.

L100 - 70 hPa is not the bottom of the stratosphere. It is actually safely in the stratosphere away from the region that can be impacted by processes around the tropopause..

>> The 70 hPa level is commonly regarded as the upper bound of the tropical tropopause layer (Fueglistaler et al., 2009, Rev. Geophys.), which could also be interpreted as the lower bound of the

tropical stratosphere. Though, the text has been adjusted to "at about 70 hPa (near the bottom of the tropical stratosphere)" in the revised manuscript [L104].

L103 Therefore, at an arbitrary altitude, the fluxes..->eastward fluxes?

>> modified: "the fluxes" → "these fluxes" [L107]

L104..by the cease of the westerly phase - please rehprase.

>> rephrased: "cease" → "cessation" [L109]

L115..too fast to measure using monthly data.. - From the methodology, I thought that hourly data on model levels were used for this..

>> The methodology part has been clarified in the revised manuscript [L65].

L121 If the results are not sensitive to the tracking velocity, than you should choose it equally to the westerly phase for consistency.

>> The amplitude of the easterly phase is much larger than that of the westerly phase (roughly twice), and in this regard, it may be more consistent to use a larger reference wind for the easterly phase.

Eq. 2 and around L130 - You should note that eq. 2 follows Lindzen and Holton and that interested reader can find more information on its derivation and assumptions behind in this reference. Particularly, purely vertical propagation is assumed.

>> Unlike Eq. (2), Lindzen and Holton (1968) excluded the vertical advection. Therefore, their study is cited exclusively for the wave flux term of Eq. (2) in the original manuscript (the other part of the derivation does not use any specific knowledge). In addition, although LH's study assumed the pure vertical propagation, the same equation can still be utilized for obliquely propagating waves, if an effective latitudinal band is appropriately chosen for the waves driving the QBO. We use a wider latitudinal band (±15°) for the wave flux entering the stratosphere, compared to the band for the QBO wind phase (±5°), to account for some contribution of latitudinal wave propagation [L147–148].

L131..wave-forcing->wave dissipation

>> modified [L135]

L140 - Taking into account only waves <2000 km - have you tested, whether different parts of the spectrum would produce similar results?

>> Yes, I confirmed that similar regression results were produced with waves of $\lambda < 1000$ km. However, the waves with wavelengths of ~500 km (say, the dominant part in the $\lambda < 1000$ km wave spectrum) may be only marginally resolved in ERA5. It is therefore safe to use $\lambda < 2000$ km to contain a bit more spectrum. In addition, regression results using larger-scale waves ($\lambda > 2000$ km) are additionally mentioned in the revised manuscript [L199–202].

L141-142 ...wide latitude band ..for lateral propagation.. - Note that Eq. 2 and the estimation of momentum dissipated at the critical level is derived assuming vertical propagation only!

Meridional widths - the different widths for different variables bring an undesired aspect of subjectivity to the analysis. What prevents you from choosing it homogeneously across variables and base it on some objectively determined quantity as for example the width of the tropics?

>> Although Lindzen and Holton's study assumed the pure vertical propagation, the same equation can still be utilized for obliquely propagating waves with an appropriately chosen source latitude. Given the latitudinal structure of the wave flux's seasonal cycle shown in Fig. 9c, the seasonal variation of wave flux does not change much with averaging latitude band [L273], and in this reason, even the regression results were not sensitive to the averaging latitude band when a narrower band was used (not shown). Though, we keep using ±15° for the conceptual reason explained above.

L147 Again...- Again the manuscript would greatly benefit from consistency of the choices across the manuscript and QBO phases.

>> Please see my response to the comment on L121.

L149 What is the target pressure and why is the number of samples different for each pressure level?

>> The target pressure is the altitude where the regression analysis is performed. The number of samples can be explained by the number of months during which the chosen wind phase ($U_\varphi$) is at an altitude within $p \pm (1/8)p$. Therefore, it is inversely proportional to the descent speed of the phase. The text has been modified to clarify these [L155–157].

L151 Note that the sampling at the monthly interval is appropriate, given that the small-scale wave momentum flux varies largely between months. - I do not understand the logic of this sentence. Can you provide more reasoning? The flux is highly intermittent and will vary largely also between days, weeks..

>> As long as the descent-speed variation that we are interested in is on monthly or longer time scales, we do not need shorter-term variability of the wave flux but its monthly averaged variations. On the other hand, if the monthly-averaged flux variation is too slow (i.e., highly auto-correlated), using all the samples could be ineffective in terms of the degree of freedom in statistics, and one may consider even larger intervals of time series with statistically independent elements. The text has been clarified in the revised manuscript [L158–159].

L156 constant-scale height - by using the conversion to log-pressure meters, you neglect the changes in the thickness of pressure levels due to potential variations of temperature.

>> This is true, if one wants to obtain the geometric-altitude based speed rather than the speed in the log-pressure coordinates. The mean temperature ranges from ~200 K to 230 K at 70–10 hPa, which corresponds to $H = 6.3 \pm 0.4$ km. This may lead to up to ±7% differences between the geometric-altitude based speed and the log-pressure speed. However, as our equations and data are based on (log-)pressure coordinates, it is theoretically more consistent to use the constant scale height and measure the log-pressure speed.

Fig. 5 - It is not clear, whether EPFD stands for small-scale waves only. But the half scale of e) indicates a large role for the residual (or longer waves).

>> The EP flux divergence represents the total wave forcing including all waves (as represented by the dataset), without any filtering, as introduced in Eq. (1). In the revised manuscript, the figure caption now

cites Eq. (1), where it has been clarified that this term refers to total wave forcing [L91]. The half scale therefore indicates a large role of unresolved waves for ERA5, as discussed in Sect. 6.

L189 ..up to potential errors in upwelling velocity- Again, GW parameterizations and reanalysis increments play a role plus potential error of the method (assumptions, numerical implementation).

>> Please see my response to the comment on L90.

L190 The direct estimate of the wave forcing - It is not clear at this point whether it is all waves or short waves only.

>> The description has been clarified in the revised manuscript [L225; Fig. 6 caption; L91].

Fig. 8 - It clearly demonstrates the importance of UTLS winds for modulating the fluxes but also raises a question on the meridional extent of your analysis. It seems that it should be wider not to artificially cut the flux fields poleward from 15deg. Also, for wider domains, hemispheric asymmetric distribution would emerge and need to be taken into account as an important feature of the small scale forcing.

>> The figure (#9 in the revised manuscript) shows the fluxes between 20°N and 20°S. In a previous study (Kim et al., 2024), the meridional extent by which the waves travel to the equator with substantial momentum flux was modeled to be mostly less than ~15°. While we choose the ±15° band based on this [L147], the regression results (which Fig. 9 is explaining) were not sensitive to the averaging latitude with the bands of ±20° or less (not shown). Using an even wider band would change the results, but I believe it is less likely that gravity waves with substantial momentum flux propagate across such a large meridional extent (> 20°) between ~84 hPa and the regression altitude (15 hPa, at most).

Around L240 - The secondary circulation effect should be described better on a prominent place of the manuscript.

>> The effect of seasonal variability of upwelling (widely studied in the literature as a possible source of the descent speed variation) has been further discussed in the revised manuscript, synthetically along with the wave flux, in the newly added Fig. 5 [L176–193]. The related statement has also been added in Abstract [L5] and in Sect. 5 [L281–282]. In addition to this, the effect of mean upwelling remains included, as in the original manuscript. However, the effect of QBO-induced local circulation is not addressed in this study, as (1) it depends solely on the QBO phase, by definition, which is fixed in our descent speed analysis at each altitude, and (2) it is considered a response of the wave–mean flow interaction of the QBO rather than a direct driver of the QBO variability.

L256 - gravity-wave momentum flux estimated in the tropical tropopause layer. - This would be desirable. But, you analyze GW flux at 70hPa, i.e. safely above TTL after it already interacted with the winds here.

>> Please see my response to the Major comment.

Around L285 Data availability - More effort should be done towards soliciting the reproducibility. For example the datasets and time series used to produce the figures and regression should be made available or at least the algorithms for the methodology.

>> This part has been improved in the revised manuscript.

---

## Author Response (AR3)

**Responses to Editor's Comments**

I would like to thank you for preparing the revised version and mostly reflecting the referee and my comments. I think that the manuscript has been improved substantially and I am particularly happy that your results hold also for the fluxes at 84hPa.

Based on my own quick assessment I identify a few minor remaining points that should be clarified before the publication. Please read them as a set of unbinding editorial suggestions.

>> I appreciate the editor's comments with careful attention to the manuscript. During this revision, we have additionally carried out the analysis suggested originally by Referee #3 and the editor, and included a new section discussing its results. We have also made effort to clarity the points raised by the editor.

1) The comment from Ref#3 regarding.. Alternatives to this correlational analysis are possible. For example, I expected that the paper was
going to take advantage of the descent rate formalism in Equation 2 to formulate a quantitative descent rate budget...
I think that this is a very valid comment and providing a quantitative proof of your hypothesis would be superior to the correlation analysis.
I see that this would practically mean a whole new analysis, but, I think that it would be great, if you add a few lines in the conclusions, where you recommend alternative ways how your hypothesis can be further examined and mention this suggestion by the referee. (As you present a new paradigm shifting hypothesis, you must expect that there will be papers testing it in the future and you should support this process).

>> In the revision, this analysis suggested previously by Referee #3 has been performed and its results have been added in the revised manuscript, with new Fig. 9 and Table 1 and the corresponding text in a new section (4.3). Please see Sect. 4.3 for the results.

2) The issue of indirect estimate-
>> Equation (1) is the generally applied momentum equation, not specific to reanalyses (no external forcing from unresolved/parameterized processes or data assimilation is introduced on its RHS). Therefore, the EP flux in Eq. (1) accounts for the total wave forcing from waves of all scales. We have
clarified this in the revised manuscript [L91–92]. The indirect estimate, therefore, represents the same quantity...

Of course that Eq. 1 is general and does not include the processes that I mentioned. But, you have to acknowledge that you are applying it on reanalysis data, where the zonal mean momentum tendency is affected also by the parameterized tendencies and analysis increments. Further, the fact that you are diagnosing the terms on a non-native grid and output sampling results in possible inaccuracy of your computation that would also project to the indirect estimate. Please provide a detailed reasoning, why you think that I am wrong here and reflect this clearly in the manuscript.

>> We fully agree that the wind tendency in reanalysis includes contributions from parameterized tendencies and analysis increments. However, our study does not attempt to separate or quantify these individual contributions. Rather, we apply the momentum equation as written and calculate the total wave forcing indirectly as the residual between the wind tendency and the remaining term (Adv). In this framework, the indirect estimate formally reflects the total wave forcing as defined by the governing

equation, without further consideration of how the wind tendency is produced within the reanalysis system. Any contributions from parameterizations or data assimilation are inherently included in the residual and are not treated as separate sources.

As noted in the manuscript [L222], we acknowledge that uncertainty in the indirect estimate arises from potential errors in Adv, by construction. Since the purpose of this study is not to analyze reanalysis-specific processes or their attribution, we respectfully maintain that the current level of explanation is sufficient.

3) In your responses you made the case for different latitudinal averaging and cut-off utilization throughout the manuscript. Please reflect this in the manuscript, where applicable, to make any subjective choice you made absolutely transparent for the interested readers.

>> We have reviewed the latitudinal averaging and cut-off values specified in the manuscript and confirmed that the reasoning behind these choices are clearly stated.

- The latitude band for the stratospheric fluxes in Fig. 1 has been changed to 5°N–5°S (matching with the winds). We realized during this revision that the consistent averaging with the winds is more appropriate here, as this figure focuses on fluxes at stratospheric levels where the QBO is forced (cf. flux entering the stratosphere). Thank you for the comment. Now, the wider latitude band is used only for the fluxes entering the stratosphere (at 84 hPa) or lower.

- Regarding the cut-off value, we have added an additional justification for our choice [L154].

L71 I reiterate that 1-2-1 smoothing definition needs to be provided (or give a reference here).

>> It is now further clarified: "the three-point moving average with coefficients [1, 2, 1]/4"

L136 and eq. 2 - provide a reference or an analytical proof that the vertical propagation assumption can be relaxed when using the Lindzen and Holton approximation for the EPFD in the presence of critical layers.

>> Lindzen and Holton's derivation of momentum forcing due to the critical-level absorption relies essentially on the dimensionality of wind shear rather than that of wave propagation. If the variation in the ambient flow along the wave path in the critical layer is dominated by the vertical change of the flow (namely, its meridional change is relatively ignorable), then the situation is the same as in LH's case, even for oblique wave propagation (except the path of waves). In this context, no additional derivation is required beyond that of LH. We have now clarified in the revised manuscript that we consider the critical-level absorption under vertical shear [L134–135].

L201 and other (not shown) statements in the manuscript - Consider presenting these results in the Appendix.

>> We have carefully reviewed all instances of "(not shown)" in the manuscript during this revision.

- The paragraph containing L201 (on the limited contribution of larger-scale waves) has been removed, since this content is now addressed more effectively in the newly added Sect. 4.3 (in response to Comment #1) [L280–282]. The analysis in this section examines the contribution of momentum forcing exerted locally on the descent speed, providing a more quantitative view than the correlation-based regression analysis done in Sect. 4.1. The larger-scale momentum forcing and its seasonal variation are much smaller (deduced from older reanalyses with less small-scale waves being resolved; Figs. B1–B2) compared to those seen in Figs. 6–8 and 9. This may provide a stronger argument for the limited role of

large-scale waves than the original discussion around L201 (which was based on the regression performance alone).

- The eastward momentum fluxes due to small-scale waves are now presented in Fig. C1 (cf. Fig. 10).

- We have chosen to retain the remaining "(not shown)" statements, as the omitted results are rather too detailed and would not substantially benefit the clarity or focus of the manuscript.

Code and data availability - Although this is not yet strictly required by ACP, consider publishing your code in some public repository (Zenodo). As I wrote above, you present a new paradigm shifting hypothesis and you must expect and solicit further work on testing the hypothesis.

>> I appreciate the suggestion to make the analysis code publicly available, and fully agree that this practice can facilitate transparency and reproducibility. However, the current scripts were written for internal use and are not yet in a form suitable for public release. Preparing and documenting them for external use would require substantial time and effort.

At this stage, I have chosen to make the scripts and data available upon request, as noted in the manuscript. I hope this will still support further work and collaboration, and I am open to sharing specific parts of the code with interested researchers if needed.